**Drainage reorganization and divide migration induced by the excavation of the Ebro**
**basin (NE Spain)**
Arnaud Vacherat [1], Stéphane Bonnet [1], Frédéric Mouthereau [1]
[1]Géosciences Environnement Toulouse (GET), Université de Toulouse, CNRS, IRD, UPS,
(Toulouse), France
*Correspondance to*: Stéphane Bonnet (stephane.bonnet@get.omp.eu)
**Abstract**
Intracontinental endorheic basins are key elements of source-to-sink systems as they preserve
sediments eroded from the surrounding catchments. Drainage reorganization in such a basin in
response to changing boundary conditions has strong implications on the sediment routing
system and on landscape evolution. The Ebro and Duero basins represent two foreland basins,
which developed in response to the growth of surrounding compressional orogens, the Pyrenees
and the Cantabrian mountains to the north, the Iberian Ranges to the south, and the Catalan
Coastal Range to the east. They were once connected as endorheic basins in the early Oligocene.
By the end of the Miocene, new post-orogenic conditions led to the current setting in which the
Ebro and Duero basins are flowing in opposite directions, towards the Mediterranean Sea and
the Atlantic Ocean. Although these two hydrographic basins recorded a similar history, they are
characterized by very different morphologic features. The Ebro basin is highly excavated,
whereas relicts of the endorheic stage are very well preserved in the Duero basin. The
contrasting morphological preservation of the endorheic stage represents an ideal natural
laboratory to study the drivers (internal / external) of post-orogenic drainage divide mobility,
drainage network and landscape evolution. To that aim, we use field and map observations and
we apply the χ-analysis of river profiles along the divide between the Ebro and Duero drainage
basins. We show here that the contrasting excavation of the Ebro and Duero basins drives a
reorganization of their drainage network through a series of captures, which resulted in the
southwestward migration of their main drainage divide. Fluvial captures have strong impact on
drainage areas, fluxes, and so on their respective incision capacity. We conclude that drainage
reorganization driven by the capture of the Duero rivers by the Ebro drainage system explains
the first-order preservation of endorheic stage remnants in the Duero basin, due to drainage area
loss, independently from tectonics and climate.
**1. Introduction**
Landscapes subjected to contrasted erosion rates between adjacent drainage basins show a
migration of their drainage divide toward the area of lower erosion rates (Bonnet, 2009; Willett
et al., 2014). This is the case for mountain ranges characterized by gradients in precipitation
rates due to orography, once landscapes are in a transient state and are not adjusted to
precipitation differences (Bonnet, 2009). It also occurs when drainage reorganized in response
to capture (Yanites et al., 2013; Willett et al., 2014). River capture actually drives a drop in the
spatial position of drainage divide (Prince et al. 2011) but also produces a wave of erosion in
the captured reach (Yanites et al., 2013) that may impact divide position. Historically, migration
of divides has been inferred by changes in the provenance of sediments stored in sedimentary
basins (*e.g.* Kuhlemann et al., 2001). It is however a process that is generally very difficult to
document in erosional landscapes. Recent developments have provided models and analytical
approaches to identify divide migration in the landscape (Bonnet, 2009; Castelltort et al., 2012;
Willett et al., 2014; Whipple et al., 2017). Among them the recently-developed χ-anaysis of
longitudinal profiles of rivers (Perron and Royden, 2012) is based on the recognition of
disequilibrium along river profiles, disequilibrium being defined by the departure from an ideal
equilibrium shape. The application of this method to both natural and numerically-simulated
landscapes, has allowed to demonstrate contrasts in the equilibrium state of rivers across divide
and then to infer their migration (Willett et al., 2014). The applicability of this method is
however limited to settings where the response time of rivers is larger compared to the rate of
divide migration, so they can actually show disequilibrium in their longitudinal profiles
(Whipple et al., 2017).
The Ebro and Duero drainage basins in the Northern Iberian Peninsula show geological and
geomorphological evidence of very contrasted erosional histories during the Neogene. They
initially recorded a long endorheic stage from the Early Oligocene to the Late Miocene (Riba
et al., 1983; Garcia-Castellanos et al., 2003). Since then, both basins opened toward the Atlantic
Ocean (Duero) or the Mediterranean Sea (Ebro). The Ebro basin's opening is reflected in the
landscape by evidence of river incision (Garcia-Castellanos et al., 2003), whereas the Duero
Basin does not show significant incision in its upstream part as a large relict of its endorheic
morphology is preserved (Antón et al., 2012). The Duero river long profile actually shows a
pronounced knickpoint (knickzone) defining an upstream domain of high mean elevation (~800
m) and low relief where the sediments deposited during the endorheic stage are relatively well
preserved. Then, these two adjacent basins are characterized by differences in incision and in
the preservation of their endorheic stages. They thusrepresent an ideal natural laboratory to
evaluate  divide migration in response to differential post-orogenic incision . Following a
presentation of the geological context, we first compile evidence of fluvial captures along the
Ebro-Duero divide, based on previous studies and our own investigations, and we map the
location of knickpoints and relict portions of the drainage network. We use all these
observations to reconstruct a paleo-divide position and to estimate the impact of divide
migration in terms of drainage area and stream power. We complement this dataset by providing
a map of $\chi$ across divide (Willett et al., 2014) to highlight potential disequilibrium state between
rivers of the Ebro and Duero catchments.
**2. Geological setting**
2.1 The Ebro and Duero basins
The Ebro and Duero basins represent two hydrographic basins located in the northern part of
the Iberian Peninsula (Fig. 1). The bedrock of the Ebro and Duero drainage basins mainly
consists of Cenozoic deposits, and Mesozoic and Paleozoic rocks in their headwaters (Fig. 2).
They formed once a unique foreland basin during the Cenozoic controlled by the flexural
loading by the surrounding mountain belts: the Pyrenees and the Cantabrian mountains to the
north (Pulgar et al., 1999), the Iberian and Central Ranges to the south (Guimerà et al., 2004;
De Vicente et al., 2007), and the Catalan Coastal Range (CCR) to the east (López-Blanco et al.,
2000 ; Salas et al., 2001), during collision between Iberia and Europe since the Late Cretaceous.
From the Late Cretaceous, the Ebro and Duero basins were essentially filled by clastic deposits,
and opened toward the Atlantic Ocean in the Bay of Biscay (Alonso-Zarza et al., 2002). During
the Late Eocene – Early Oligocene, the uplift in the Western Pyrenees (Puigdefàbregas et al.,
1992) led to the closure of the Ebro and Duero basins as attested by the Ebro basin
continentalization dated at ~36 Ma (Costa et al., 2010). The center of these two basins became
long-lived lakes filled with lacustrine, sandy, and evaporitic deposits from the Oligocene to the
Miocene (Riba et al., 1983; Alonso-Zarza et al., 2002; Pérez-Rivarés et al., 2002, 2004; Garcia-
Castellanos et al, 2003; Garcia-Castellanos, 2006; Larrasoaña et al., 2006; Vázquez-Urbez et
al., 2013). The opening of the Ebro basin through the Catalan Coastal Range toward the
Mediterranean Sea occurred during the Late Miocene, leading to kilometer-scale excavation
throughout the basin (Fillon and Van der Beek, 2012; Fillon et al., 2013; Garcia-Castellanos
and Larrasoaña, 2015). The exact timing and and processes driving the opening, as well as the
role of the Messinian Salinity Crisis, have long been debated (Coney et al., 1996 (post-
Messinian); Garcia-Castellanos et al., 2003 (13-8.5 Ma); Babault et al., 2006 (post-Messinian);
Urgeles et al., 2010; Cameselle et al. (2014) (Serravallian-Tortonian); Garcia-Castellanos and
Larrasoaña, 2015 (12-7.5 Ma)). In contrast with the Ebro basin, incision in the upper Duero
basin appears much less significant. The Duero basin is characterized by a low relief topography
(Fig. 1) in its upstream part, at 700-800 m above sea level to the west, and at 1000-1100 m a.s.l.
to the north, northeast, and to the east in the Almazan subbasin, close to the divide with the
Ebro basin. The connection of the Duero River with the Atlantic Ocean occurred from the Late
Miocene-Early Pliocene to the Late Pliocene-Early Pleistocene (Martín-Serrano, 1991). The
current Ebro and Duero drainage networks are separated by a divide running from the
Cantabrian belt to the NW, toward the SE in the Iberian Range (Figs. 1, 2, 3). In the following,
we review the geological evolution of the different domains that constitute this drainage divide
between the Ebro and Duero drainage basins.
2.2 The Iberian Range
The Iberian Range (Figs. 2, 4) is a double vergent fold-and-thrust belt resulting from Late
Cretaceous inversion of Late Jurassic-Early Cretaceous rift basins during Iberia – Europe
convergence (Salas et al., 2001; Guimerà et al., 2004; Martín-Chivelet et al., 2002). It is divided
into two NW-SE directed branches, the Aragonese and the Castillian branches, separated by the
Tertiary Almazan subbasin (Bond, 1996). The Almazan subbasin is connected to the Duero
basin since the Early Miocene (Alonso-Zarza et al., 2002).
The Iberian Range is essentially made of marine carbonates and continental clastic sediments
ranging from Late Permian to Albian, overlying a Hercynian basement. The Cameros subbasin
to the NW represents a late Jurassic-Early Cretaceous trough almost exclusively filled by
continental siliciclastic deposits (Martín-Chivelet et al., 2002 and references therein; Del Rio
et al., 2009). Shortening in the Iberian Range occurred from the Late Cretaceous to the Early
Miocene, along inherited Hercynian NW-SE structures (Gutiérrez-Elorza and Gracia, 1997;
Guimerà et al., 2004; Gutiérrez-Elorza et al., 2002). The opening of the Calatayud basin in the
Aragonese branch occurred during the Early Miocene in response to right-lateral transpression
on the southern margin of the Iberian Range (Daroca area) (Colomer and Santanach, 1988). It
is followed during the Pliocene and the Pleistocene, by pulses of extension reactivating faults
in the Calatayud basin, and the formation of grabens such as the Daroca, Munébrega,
Gallocanta, and Jiloca grabens (Fig. 4; Colomer and Santanach, 1988; Gutiérrez-Elorza et al.,
2002; Capote et al., 2002). This is also outlined by the occurrence of Late Pliocene to Early
Pleistocene breccias and glacis levels in the Daroca and Jiloca grabens (Gracia, 1992, 1993a;
Gracia and Cuchi, 1993; Gutiérrez-Santolalla et al., 1996). These Neogene troughs are filled by
continental deposits and pediments, up to the Quaternary (Fig. 4). The Neogene tectonic pulses
in the Iberian are interrupted by periods of quiescence during which erosion surfaces developed
(Gutiérrez-Elorza and Gracia, 1997).
Deformation and uplift of the Iberian Range and Cameros basin resulted in the development of
a new drainage divide between the Duero and Ebro basins and in the isolation of the Almazan
subbasin (Alonso-Zarza et al., 2002). In contrast, the connection between the Duero the Ebro
basins has not been affected by significant deformation and uplift in the proto-Rioja trough
(Mikes, 2010).

2.3 The Rioja trough and  Bureba high

The Rioja trough (Figs. 2, 5) recorded important subsidence, especially during the Cenozoic (>
5 km), related to compression and thrusting on its borders (Jurado and Riba, 1996). As thrusting
initiated in the Pyrenean-Cantabrian belt and in the Iberian Range and Cameros basin, the Rioja
trough became domain of important synorogenic sediment transfer between the Ebro and Duero
basins. During the Paleocene, the Rioja trough was a marine depositional environment. With
the increase of sediment fluxes that originated from the exhumation of surrounding mountain
bets, sedimentation became essentially continental in the Eocene. Thrusting continued during
the Oligocene resulting in the formation of an anticline connecting the Cantabrian domain and
the Cameros inverted basin. This morphologic high (the Bureba anticline, Fig. 5) located in the
center of the area is supposed to have triggered the disconnection between the Duero and Ebro
basins (Mikes, 2010), as suggested by the repartition of alluvial fans on both sides of this
structure (Muñoz-Jiménez and Casas-Sainz, 1997; Villena et al., 1996). During the Miocene,
deformation ceased as evidenced by the deposition of undeformed middle Miocene to Holocene
strata. The Bureba anticline is cored by Albian strata and topped by Santonian limestones and
Oligocene conglomerates controlling the location of the current main drainage divide between
the Ebro and Duero river networks (Fig. 5).

2.4 Climate evolution

Climate exerts a major control on valley incision, sediment discharge, and on the evolution of
drainage networks (Willet, 1999; Garcia-Castellanos, 2006; Bonnet, 2009; Whipple, 2009;
Whitfield and Harvey, 2012; Stange et al., 2014). The mean annual precipitation map for the
North Iberian Peninsula (Hijmans et al., 2005) shows a similar pattern for both the Ebro and
Duero basins as they record very low precipitation, associated with global subarid conditions,
with the exception of the Cameros basin that record a slightly higher precipitation rate (Fig. 6).
There is a strong contrast to the north, toward the Mediterranean Sea and the most elevated
areas in the Cantabrian and Pyrenean belts, where precipitation drastically increases.
The paleoclimatic evolution from the Late Cretaceous to the Neogene is linked both with the
effects of surrounding mountains uplift, and with the latitudinal variation drift of Iberia from
30°N in the Cretaceous to ~40°N during Late Neogene times. The hot-humid tropical climate
of the Late Cretaceous became drier and arid from the Paleocene to the Middle Miocene (López-
Martínez et al., 1986), favouring the development of endorheic lakes (Garcia-Castellanos,
2006). During the Middle-Late Miocene and Early Pliocene, the northern Iberia recorded more
humid and seasonal conditions (Calvo et al., 1993; Alonso-Zarza and Calvo, 2000) with
alternations of cold-wet and hot-dry periods (Bessais and Cravatte, 1988; Rivas-Carballo et al.,
1994; Jiménez-Moreno et al., 2010). More humid and colder conditions took place in the Late
Pliocene, characterized by dry glacial periods and humid interglacials (Suc and Popescu, 2005;
Jiménez-Moreno et al., 2013). Climatic contrasts increased, triggering intense glaciers
fluctuations in the surrounding mountain ranges during the Lower-Middle Pleistocene transition
(1.4-0.8 Ma) (Moreno et al., 2012; Duval et al., 2015; Sancho et al., 2016), and throughout the
Late Pleistocene period, which record glacial / interglacial oscillations, as evidenced by pollen
identification (Suc and Popescu, 2005; Jiménez-Moreno et al., 2010, 2013; Barrón et al., 2016;
García-Ruiz et al., 2016) and speleothem studies (Moreno et al., 2013; Bartolomé et al., 2015).
Glaciers are considered as very efficient erosion tool in continental environment. They are
likely to influence drainage divide migration (Brocklehurst and Whipple, 2002). There is large
evidence of glaciers development especially for the Late Pleistocene in the Pyrenees (Delmas
et al., 2009; Nivière et al., 2016; García-Ruiz et al., 2016), in the Cantabrian belt (Serrano et
al., 2013, 2016; García-Ruiz et al., 2016), and in the Central Range (Palacios et al., 2011, 2012;
García-Ruiz et al., 2016). However, although numerous moraines have been mapped throughout
the Iberian Range (Ortigosa, 1994; García-Ruiz et al., 1998; Pellicer and Echeverría, 2004),
there is no evidence of U-shaped valleys and because of the lack of very high elevated massifs
(>2500 m), the occurrence of active ice tongues are considered as limited, if not precluded
(García-Ruiz et al., 2016).

**3. Evidence of divide mobility between the Duero and Ebro catchments**

The easternmost part of the Duero river is opposed to the Ebro tributaries that are the Jalon,
Huecha, Queiles, Alama, Cidacos, Iregua, and Najerilla rivers, whereas the Arlanzon and
Pisuerga rivers (Duero tributaries) are opposed to the Najerilla, Tiron, Oca, and Rudron rivers,
and to the westernmost part of the Ebro river (Fig. 3). The northeastern part of the Duero basin
(the easternmost Duero river, the Arlanzon and Pisuerga rivers) mainly consists of broad flat
valleys characterized by low incision depth and low-gradient streams with concave longitudinal
profiles (Antón et al., 2012, 2014). By contrast, the western part of the Ebro basin is
characterized by more incised valleys, especially in the Cantabrian and in the Cameros – Iberian
Range domains, with more complex longitudinal profiles (knickpoints, remnants of high
elevated surfaces). Previous studies (Gutiérrez-Santolalla et al., 1996; Pineda, 1997; Mikes,
2010) already shown that the Jalon and Homino rivers, which belong to the Ebro basin, have
recently captured parts of the Duero basin in the Iberian Range and in the Rioja trough,
respectively. Such evolution has been recorded by the occurrence of geomorphological markers
as wind gaps and elbows of captures, as well as by the presence of knickpoints and/or remnants
of high elevated surfaces in river long profiles. To highlight this dynamic evolution, we
performed a morphometric analysis of rivers all around the divide separating the Ebro basin
from the Duero basin, with particular attention given to the Aragonese branch of the Iberian
Range (Fig. 4) and to the Rioja Trough (Fig. 5), where captures have already been described.
The studied basins were digitally mapped using high-resolution (~30 meters) digital elevation
models (DEMs) from SRTM 1 Arc-Second Global elevation data available at the U.S.
Geological Survey (www.usgs.gov). The different DEMs were assembled using the ENVI
software. We also used 1:50,000 geological maps from the Instituto Géologico y Minero de
España (www.igme.es). We used the TopoToolbox, a MATLAB-based software developed by
Schwanghart and Scherler (2014), to extract the river network and longitudinal profiles and the
χ-analysis Tool developed by Mudd et al. (2014).

### 3.1 Fluvial captures and related knickpoints in the Iberian Range

Neogene tectonics in the Iberian range controlled the uplift of topographic ranges and the formation of several basins whose connection with the Ebro or the Duero has occasionally changed through time. Nowadays, the western part of the Almazan subbasin (Figs. 2, 4) belongs to the Duero catchment, its eastern part being drained by the Ebro drainage network and especially by the Jalon river and its tributaries (Fig. 4). Gutiérrez-Santolalla et al. (1996) proposed that the Jalon river captured this domain after cutting into the Mesozoic and Neogene strata and the two Paleozoic ridges of the Aragonese branch of the Iberian Range. They used chronostratigraphic evidence to build a relative chronology of capture events in the Jalon area. First, the incision of the northern Paleozoic ridge and capture of the Calatayud basin by the Jalon river is attributed to a post-Messinian age. The Jiloca river, the easternmost main Jalon tributary, is then thought to capture the Daroca graben area to the east during the Late Pliocene – Early Pleistocene. This is followed from the Early to Late Pleistocene by the capture of the Jiloca graben to the southeast and finally by the capture of the Munébraga graben to the southwest, by the Jalon river (Gutierrez-Santolalla et al., 1996), toward the easternmost part of the Almazan subbasin.

The Jalon river and tributaries show knickpoints in their longitudinal profiles (Fig. 4), at locations that are consistent with the events of captures proposed by Gutiérrez-Santolalla et al. (1996), suggesting that these captures are actually witnessed by knickpoints. The capture of the Jiloca graben corresponds to a major knickpoint in the Jiloca river profile that appears very smoothed, and that is followed by an upstream ~50 km long flat domain preserved at ~1000 m high above sea level. This imparts a convex shape to the Jiloca profile (Fig. 4). Due to the short period of time between the formation of the Jiloca graben (the earliest glacis deposits are attributed to the Middle Pliocene) and its capture (Early Pleistocene; Gutierrez-Santolalla et al., 1996), we suggest this upstream domain was a short-lived endorheic domain that has never been externally drained before being captured by the Ebro network. In the northwestern part of the Jiloca graben, the Cañamaria river, a tributary of the Jiloca river, heads to the northwest, reaching the Gallocanta basin, also considered as a former graben (Gracia, 1993b; Gracia et al., 1999; Gutiérrez-Elorza et al., 2002). The upstream part of its river long profile is characterized by a sharper knickpoint at the entrance of the basin, and is followed by a ~15 km long flat domain (Fig. 4). Similarly to the Jiloca graben, the Gallocanta basin appears to be a short-lived endorheic domain that has been more recently captured by the Jiloca river network.

According to Gutiérrez-Santolalla et al. (1996), the Jalon river reached the southern Paleozoic
ridge of the Aragonese branch, to the southwest of the Calatayud basin, captured the Munébrega
graben and the Almazan subbasin (also characterized by a pronounced knickpoint) during the
Pleistocene-Holocene, slightly after the capture of the Jiloca graben by the Jiloca river. This is
coherent with morphological analysis of longitudinal profiles, as the major knickpoint related
to the capture of the Jiloca graben appears very smoothed, whereas knickpoints observed in the
west are sharper, suggesting they are younger. However we cannot ruled out some local
influence of the lithology on the shape of these knickpoints.
Finally, the Piedra river (Jalon tributary) long profile shows major sharp knickpoints and two
successive ~30 km long almost flat domains in the Almazan subbasin, at ~900-1000 m above
sea level (Fig. 4). In addition, the upper reach of the river long profiles of the Jalon river, and
of its tributary the Blanco river, are characterized by major sharp knickpoints, and by a ~15 km
long flat domain at ~1000-1100 m above sea level, in the Mesozoic Castillan branch of the
Iberian Range (Fig. 4).

3.2 Fluvial captures and related knickpoints in the Rioja trough area

In the Rioja trough area, the position of the Ebro-Duero divide is partly controlled by the Bureba
anticline. It consists of folded Middle Cretaceous to Early Miocene series, covered by
undeformed Middle Miocene to Holocene deposits (Fig. 5). The anticline is orientated E-W to
the west and NE-SW to the east. The western part of the Rioja trough to the west of the NE-SW
directed branch of the Bureba anticline (Fig. 5), used to be drained toward the Duero basin since
the Oligocene (Pineda, 1997; Mikes, 2010). The westward migration of the divide to its current
location is thought to have occurred in several steps of captures as shown by the occurrence of
remnants of escarpments during the Late Miocene - Pliocene (Mikes, 2010). Once the eastern
branch of the Bureba anticline has been incised, the Ebro tributaries captured the western part
of the Rioja trough, up to the E-W branch of the Bureba anticline to the southwest, from the
Late Miocene to the Pliocene. The western part of the anticline forms a topographic ridge that
is incised by Jordan river (Fig. 5) in a place where the divide between the Ebro and Duero river
networks is located to the north of the ridge. To the East of this location however, the
topographic ridge formed by the Bureba antcline controls the current location of the main
drainage divide (Fig. 5). Here, the ridge exhibits several wind gaps, located on the northward
prolongation of the Hoz, Rioseras, and Nava Solo rivers (Figs. 5, 7). Further east, the Diablo
river does not incise the ridge and its headwater is located in the core of the eastern branch of

the Bureba anticline, the Fuente Valley (Fig. 5). These last streams are tributaries of the Ubierna river, which is a tributary of the Arlanzon river and so, of the Duero river. To the north, the Ebro river system is represented, from west to east, by the Homino river (a tributary of the Oca river) and its four tributaries, the Molina, the Fuente Monte, the Zorica, and the San Pedro rivers (Figs. 5, 7). All these streams are outlined by Late Pleistocene to Holocene alluvial series that are deposited at the bottom of their respective valleys. Valleys from the Duero side appears larger than those from the Ebro side, which are significantly more incised.

The Jordan river's headwater is located north of the ridge formed by the Bureba anticline. We can continuously follow its valley deposits northward along a broadly gentle slope, up to the locality of Coraegula (Fig. 5). However, the current course of the Jordan river is cut ~8 km south, in the vicinity of Hontomin, by the Homino (Ebro) river (Figs. 5B, C, 7). This fluvial capture is characterized by a well-defined and highly incised elbow of capture, already described by Pineda (1997) and Mikes (2010). The longitudinal profile of the Homino river shows a sharp knickpoint located on Hontomin (Fig. 7C). Finally, there is a small wind gap on the divide between the two opposite rivers (Figs. 5, 7).

To the southeast, the headwater of the Hoz river is located to the south of a wind gap cut into the Bureba ridge (Fig. 7C). To the north, in the exact prolongation of the Hoz river, the Molina river shows a bend similar to the elbow of capture previously described for the Homino river (Fig. 7) and there is a minor knickpoint located on this elbow, according to the extracted river long profile. Thus, it is likely that the Molina river used to represent the former upper reach of the Hoz river, in a period when the Ebro-Duero divide was located northward, before being captured by the Ebro network.

To the east, the Rioseras and the Nava Solo rivers have also their headwater located to the South of wind gaps in the Bureba ridge (Fig. 7). Similarly, in their exact prolongations, the Fuente Monte and the Zorica rivers show important elbows of capture with minor knickpoints. They may also represent former upper reaches of Duero streams that have been captured by the Ebro network (Figs. 5, 7, 8).

Further east, the headwater of the Diablo river is located on the depression represented by the core of the eastern branch of the Bureba anticline, the Fuente valley. In its prolongation to the northeast, the San Pedro river incises the northeastern termination of the anticline from the north before entering the valley, leading to a southward retreat of the divide (Fig. 5). Capture is again evidenced by important incision contrast between Ebro and Duero systems, and by sharp knickpoints on the upper reach of the San Pedro river long profile when crossing the Santonian dolomites (Fig. 8). According to this whole set of observations, and in agreement with previous

findings of Pineda (1997) and  Mikes (2010), we propose that the western part of the Rioja
trough, in the Bureba area has been recently captured by the Ebro drainage network leading to
a sequence of  southwestward retreat of the main drainage divide, toward the Duero basin (Fig.
7E).

A similar capture pattern can be observed further west in the continuity of the Bureba anticline
(Fig. 5). The San Anton river shows a well-defined elbow of capture accompanied by a
smoothed knickpoint (See Fig. S1 in the Supplement) at its junction with the Rudron river (Ebro
tributary). The river course is highly incised toward the east, along the northern flank of the
WNW – ESE anticline, almost connecting to the upper reach of the Ubierna river. Valley
deposits are also observed in the continuity of the Ubierna valley, which former route is
suggested by a wind gap (Fig. 5). However, this domain is no longer connected to its network
as it is now wandered from the North by the Nava river, a tributary of the Moradillo river, which
is a tributary of the Rudron river. This domain clearly records captures leading to divide
migration toward the Duero, also in favor of the Ebro basin.

3.3 Past position of the Ebro-Duero divide and implication for stream-power of the Duero River

We used all observations that support divide migration in the Iberian Range and Rioja trough
to estimate a paleo-position of the drainage divide between the Duero and Ebro drainage basins
(Fig. 9). For this purpose, we considered the location of major knickpoint along the rivers where
fluvial captures are defined. Both the Ebro river and several tributaries show high elevated ~10-
20 km long flat domains at ~800 – 1200 m a.s.l. and major knickpoints in the upper reach of
their long profiles as the Rudron, Queiles, and Alama rivers, as well as the Homino river and
its tributaries: the Puerta Nogales and Valdelanelala rivers (Figs. 5, 8; Fig. S1). All these flat
domains may not be related to surface uplift as they are not clearly associated with active
tectonic features. The Duero basin being characterized by a high mean elevation (~1000 m) and
by a very limited incision in the vicinity of the Ebro/Duero drainage divide, a sudden divide
migration toward the Duero basin is then expected to isolate such high elevated and relatively
preserved surfaces. We suggest these flat domains have been recently captured by Ebro
tributaries, and represent remnants of Duero drainage areas, integrated into the Ebro catchment
from divide retreat toward the Duero basin. Overall, we consider a paleodrainage divide
delimited by these high-elevated knickpoints and flat domains, except for the Jiloca graben area
to the southeast, characterized by the occurrence of short-lived endorheic domains (Fig. 9).
Incision in the Ebro basin leads to the capture of new drainage areas, whereas the Duero basin
recorded important loss of its own surface. The present day drainage area of the Cenozoic Duero
basin, upstream of the major knickzone observed to the west in the Iberian Massif is ~63000
km$^2$. We used the paleo-divide position shown in Figure 9 to define a « recent » captured area
that used to belong to the Duero basin. This area represents ~7700 km$^2$, which corresponds to
~12% of the present-day Cenozoic Duero basin drainage area. Such a reduction of the drainage
area could have strong implications on the evolution of the Duero basin, as important lowering
of water and sediment fluxes, and so of incision throughout the basin. To better resolve the
impact of such drainage area reduction on incision capacity, we perform a stream power analysis
of the Duero river. We consider the specific stream power, $\omega$, defined as $\omega = \rho\, g\, Q\, S\, /\, W$, where
$\rho$ is water density, $g$ is gravitational acceleration, $Q$ is discharge, $S$ is local river gradient, and
$W$ is river width (see the Supplement for details of the calculation). We calculate $\omega$ for the
present-day Duero river, and for a restored ancient Duero river that drained this 12% of lost
area. We plot the difference (ancient – present day) between the two curves in Figure 10, with
the Duero river long profile. Calculated difference in specific stream power values are relatively
low ($< 2$ W m$^{-2}$) for the upstream part of the basin, but increase to ~5 W m$^{-2}$ when approaching
the major knickzone at a distance of ~350 km from the river mouth. The knickzone is
characterized by peak values exceeding 10 W m$^{-2}$, which rapidly decrease to ~0 W m$^{-2}$ at the
base of the knickzone (~200 km) and up to the river mouth (Fig. 10). Some alternating peak
and null values are observed in the lower reach of the river and may be related to the occurrence
of numerous dams along the river. Overall, the specific stream power calculated for the ancient
Duero river show higher values than for the present day from the base of the knickzone to the
uppermost reach of the river (Fig. 10). This implies a general decrease of the Duero river's
incision capacity between this ancient state to the present day, magnified on the knickzone.

3.4 $\chi$ map

The comparison of the shape of longitudinal profiles of rivers across divide is a way that has
been proposed recently to infer disequilibrium between rivers and the potential migration of
their divide (Willett et al., 2014). The $\chi$-analysis of river profiles (Perron and Royden, 2012)
is a powerful tool to evidence differences in the equilibrium state of rivers across divide, and
then to infer their migration (Willett et al., 2014).  . This method is based on a coordinate
transformation allowing linearizing river profiles (Perron and Royden, 2012). Considering
constant uplift rate (U) and erodibility (K) in time and space the χ-transformed profile of a river
is defined by the following equation (Perron and Royden, 2012; Mudd et al., 2014):

$$z(x) = z_b(x_b) + \left(\frac{U}{KA_0^m}\right)^{1/n} \chi$$


with
$$\chi = \int_{x_b}^{x} \left(\frac{A_0}{A(x)}\right)^{\frac{m}{n}} dx$$

where z (x) is the elevation of the channel, x is the longitudinal distance, $z_b$ is the elevation at
the river's base level (distance $x_b$), A is the drainage area, $A_0$ is a reference drainage area, and
exponents m and n are empirical constants.

When using the χ variable instead of the distance for plotting the elevation z along channel, (χ-
plot), the longitudinal profile of a steady-state channel is shown as a straight line (Perron and
Royden, 2012). Any channel pulled away from this line is in disequilibrium and is then expected
to attempt to reach equilibrium. Mapping χ on several watersheds and comparing χ across
drainage divides is then a potential way to high disequilibrium between rivers across divide and
to elucidate divide migration and drainage reorganization through captures (Willett et al., 2014).
We used the χ-analysis Tool developed by Mudd et al. (2014) to select the best m/n ratio by
iteration (Perron and Royden, 2012) and to calculate χ for rivers throughout the divide between
the Ebro and Duero basins from a similar base level at 850 m a.s.l. The best mean m/n ratio for
all our streams is 0.425, which falls in the typical range of values observed for rivers (~0.4 –
0.6: e.g. Kirby and Whipple, 2012). The resulting map (Fig. 11) shows χ values calculated on
different opposite streams in the vicinity of the Ebro/Duero drainage divide. Similar values on
both sides of the divide suggest the two opposite streams are at equilibrium, whereas strong
contrasted χ values imply disequilibrium leading to divide migration, continuously or through
fluvial capture, toward the high χ values (Willett et al., 2014).The map of χ values actually
shows significant contrasting values across the Ebro/Duero divide. We comment here these
contrasts along the divide from the SE to the NW of the area considered (Fig. 11).
There is a strong contrast in χ values between the headwater of the Jalon river (Fig. 11),
characterized by low values (~300 m), and the closest part from the divide of the Bordecorex
river (Fig. 4), a tributary of the Duero river (~500 m). Such a disequilibrium implies divide
migration toward the Duero basin, predicting the capture of the uppermost reach of the
Bordecorex river by the Jalon river. To the north, tributaries of the Jalon river show slightly
lower χ values than the tributaries of the Duero river. This suggests a relative stable situation
although small captures may occur toward the Duero basin. A higher contrast is observed
around the easternmost part of the Duero basin, which is surrounded by the Ebro basin. The
Araviana river (tributary of the Duero river) seems to be taken in a bottleneck between the
Manubles river to the south and the Queiles river to the north (Fig. 4), which both show lower
χ values (Fig. 11). Toward the east, there is a strongest χ values contrast between headwaters
of the Araviana river (>700 m) and of the Isuela (Jalon tributary) and Huecha rivers (<100 m).
This domain appears clearly in disequilibrium and is expected to be captured by the Ebro
drainage network. Such high χ values differences appear also to the northwest (Fig. 11), in the
southern part of the Cameros basin where the Duero river and its tributaries' headwaters show
χ values >500-700 m, whereas the facing rivers (Alama, Cidacos, Iregua, and Najerilla) are all
characterized by low χ values <100 m. This predicts important disequilibrium and divide
migration and fluvial captures toward the south. Northwestward, χ values between Duero and
Ebro network are more similar indicating that the divide is relatively more stable here, up to the
westernmost part of the Ebro basin (Fig. 11). However, there are some slight localized χ value
contrasts (~200 / ~450 m) as observed between the Tiron and the Arlanzon rivers, between the
Rudron and the Ubierna and Urbel rivers, and between the Ebro and the Pisuerga rivers (Fig.
11). It suggests minor local captures toward the Duero basin.

To sum up, χ values calculated in the vicinity of the drainage divide between the Ebro and
Duero river networks show a general disequilibrium (Fig. 11) as the Ebro network is
characterized by low χ values (up to ~200-300 m) compared to those for the Duero network
(up to ~450-700 m). In complement with all the evidence of divide displacements induced by
captures described previously this allows predicting a general divide migration toward the
Duero basin through headwater retreat, in favor of the Ebro tributaries, especially around the
Almazan subbasin, which is expected to be entirely captured by the Ebro basin.

**4. Discussion**

4.1 Long term trend of divide migration

The oldest capture evidence in our study area corresponds to the incision of the northern part
of the Iberian Range by the Jalon river and by the capture of the Calatayud basin, attributed to
the post-Messinian (Gutiérrez-Santolalla et al. 1996). We propose, based on morphological
evidence (Fig. 4) and in agreement with stratigraphic data (Gutiérrez-Santolalla et al. 1996),
that the Jalon river system captured the Jiloca graben to the east since the Early Pleistocene,
before progressively capturing the Almazan subbasin toward the west in the Holocene
(Gutiérrez-Santolalla et al. 1996). From $\chi$−analysis (Fig. 11), we deduce that the eastern part of
the Duero basin, the Almazan subbasin, is being actively captured by Ebro tributaries that
drained the Iberian Range and the Cameros basin. Despite low contrasts in $\chi$ values, local
captures are also suggested in the vicinity of the Ebro / Duero drainage divide toward the
northwest. Capture is further implied by the occurrence of numerous high elevated (~1000 m)
knickpoints and low-relief surfaces (Figs. 5, 8, 9, 11).
Thus, there is a good correlation between $\chi$ evidence and morphological and stratigraphic data
implying long-lasting captures and divide migration during Pliocene, Pleistocene, and
Holocene times in favor of the Ebro basin.

The pursuit of such a long-term capture trend may be driven by tectonic and/or climatic forcing
(Willett, 1999; Montgomery et al., 2001; Sobel et al., 2003; Sobel and Strecker, 2003; Bonnet,
2009; Whipple, 2009; Castelltort et al., 2012; Kirby and Whipple, 2012; Goren et al., 2015; Van
der Beek et al., 2016). However, such long-term trend in drainage reorganization may also occur
in tectonically quiescent domains, independently of external forcing (Prince et al., 2011). Here,
the Iberian Range and the Cameros basin recorded extension pulses from the Late Miocene to
the Early Pleistocene, responsible for the formation of several grabens as previously described
(Gutiérrez-Santolalla et al., 1996; Capote et al., 2002). Extension events are also recorded
during the Holocene, nevertheless, the youngest erosion surface of Late Pliocene-Early
Pleistocene age observed in our study area shows no tectonic-related deformation and
reworking, suggesting that tectonic activity is reduced here (Gutiérrez-Elorza and Gracia,
1997). This is also consistent with the relative scarcity of seismic activity observed in our study
area, compared, for instance, to the Pyrenees, or to the Betics (Herraiz et al., 2000; Lacan and
Ortuño, 2012). We consequently propose that local tectonic activity is not the main driver of
the capture histories documented here, as most capture events postdate the cessation of tectonic
activity, and occur during periods of quiescence (Gutiérrez-Santolalla et al., 1996).

The Cameros Massif if characterized by relatively high mean annual precipitation up to ~1000
mm/an (Fig. 6) with high elevation (~1400-2200 m) in comparison with the surrounding areas.
This contrasts with the adjacent Ebro and Duero basins where low precipitation rates, of ~400-
500 mm/an (Hijmans et al., 2005), illustrate subarid climate conditions. The Cameros area is
the only place in our study area where a contrast in precipitation pattern (Fig. 6) would
potentially drive a migration of the divide toward the drier, Duero area. Given that the same
pattern is observed everywhere, even where there isn't any precipitation difference, we suggest
that the present day climatic condition is unlikely to control the general pattern of current
drainage reorganization between the Ebro and Duero basins. During the Pliocene and the
Pleistocene, the climatic record in the northern Iberia Peninsula is characterized by alternations
between similar subarid conditions and intense glaciation. Paleoclimate proxies do not allow to
highlight past precipitation differences along the divide that could explain past drainage
reorganization. Moreover, there is no clear evidence of important glacier development and
related erosion in our study area, especially for the Cameros basin and the Iberian Range
(Ortigosa, 1994; García-Ruiz et al., 1998, 2016; Pellicer and Echeverría, 2004). This indicates
that drainage evolution between the Ebro and Duero basins is unlikely to be related to climatic
evolution.
4.2 Excavation of the Ebro basin as the main factor controlling divide migration and limiting
incision of the Duero river
A striking morphological feature for river capture in our study area is the important contrast in
the incision pattern (e.g. Fig. 1B) from one side of the divide to the other. This suggests that the
incision capacity of the river network is the main driver for capture and divide migration. Both
tectonic and climatic forcing does not appear to control drainage reorganization between the
Ebro and Duero basins.
The opening of the Ebro basin toward the Mediterranean Sea during the Late Miocene led to
widespread excavation (Garcia-Castellanos et al., 2003, Garcia-Castellanos and Larrasoaña,
2015), favored by more humid and seasonal climatic conditions (Calvo et al., 1993; Alonso-
Zarza and Calvo, 2000). By contrast, incision related to the opening of the Duero basin toward
the Atlantic Ocean is concentrated to the west in the Iberian Massif, characterized by a
largescale knickzone (150 km long and 500 m high) in the Duero river long profile (Fig. 1B).
This contrasts with the limited eastward propagation of incision in the Cenozoic part of the
basin (Antón et al., 2012, 2014), despite climatic conditions similar to the Ebro basin. An
explanation resides in the fact that the resistant Iberian Massif basement rocks may have
controlled and limited incision and drainage reorganization in the Cenozoic Duero basin (Antón
et al., 2012). The Duero profile upstream of this major knickzone may be considered as a high
elevated local base level for its tributaries there. Difference between the Ebro and Duero base-
levels implies a major contrast in fluvial dynamics. We suggest the systematic and long-term
trend of divide migration toward the Duero basin and fluvial capture in favor of the Ebro basin
is driven by the differential incision behavior, controlled by base-level difference.
Our stream power analysis along the Duero river (Fig. 10) shows that the difference in drainage
area of the Duero inferred from our paleo-divide map (Fig. 9) induces a noticeable decrease of
stream power values of the Duero in the vicinity of the knickzone. This stream power is a
minimum estimate because calculation does not take into account possible captures and divide
migration in other areas along the Duero basin divide, nor the full history of the divide migration
through time and the related ongoing decrease in water discharge as documented in laboratory-
scale landscape experiments (Bonnet, 2009). Some contrasts of incision are also observed in
the Iberian Range along the southern border of the Duero, and in the Cantabrian domain to the
North. Both show more important incision than in the Duero basin, suggesting potential river
captures and divide migration at the expense of the Duero basin, increasing the total of lost
drainage area.  Even if it gives minimal estimate, our stream power analysis suggests that
drainage area reduction may have limited the erosion in the Duero basin. This provides an
explanation for the preservation of the lithologic barrier to the west, along the main knickzone
of the Duero considered as an intermediate, local base level (Antón et al., 2012).We propose
that the reduction of the Duero drainage area caused by captures and incision in the Ebro basin,
is responsible for a significant decrease of the incision capacity in the Duero basin. We infer
that the ongoing drainage network growth in the Ebro basin may be responsible for the current
preservation of large morphological relics of the endorheic stage in the Duero basin.
The opening of the Ebro basin toward the Mediterranean Sea resulted in a drastic base level
drop. This results in the establishment of an upstream-migrating incision wave that propagates
to every tributary of the Ebro network, responsible for knickpoints migration (Schumm et al.,
1987; Whipple and Tucker, 1999; Yanites et al., 2013) and for drainage reorganization and
divide migration. The χ-analysis that we performed along the current Ebro-Duero divide (Fig.
11) highlights areas where geomorphic disequilibrium is still ongoing, which suggests that they
are areas where divide is currently mobile. The modelling study performed by Garcia-
Castellanos and Larrasoaña (2015) suggests that the re-opening of the Ebro basin occurred
between 12.0 and 7.5 Ma. This indicates that the growth of the drainage network of the Ebro

basin and the establishment of new steady-state conditions is a long-lived phenomenon, which is still not achieved today.

**Conclusion**

In this paper we present a morphometric analysis of the landscape along the divide between the Ebro and Duero drainage basins located in the northern part of the Iberian Peninsula. This area shows numerous evidence of river captures by the Ebro drainage network resulting in a long-lasting migration of their divide, toward the Duero basin. Although these two basins record a similar geological history, with a long endorheic stage during Oligocene and Miocene times, they show a very contrasted incision and preservation state of their original endorheic morphology. Since the Late Miocene, the Ebro basin was opened to the Mediterranean Sea and record important erosion. On the opposite, the Duero was opened to the Atlantic Ocean since the Late Miocene – Early Pliocene but its longitudinal profile exhibits a pronounced knickpoint, which delimits an upstream domain of low relief and limited incision, likely representing a relict of its endorheic topography. We propose that this contrast of incision is the main driver of the migration of divide that we document. The morphological analysis of rivers across the divide highlights areas where geomorphic disequilibrium is still ongoing, which suggests that the Ebro-Duero divide is currently mobile. The quantification of the decrease of the drainage area of the Duero based on the reconstruction of a paleo-position of the Ebro-Duero divide shows that the divide migration results in a significant lowering of the stream power of the Duero river, particularly along its knickzone. We suggest that divide migration induces a decrease of the incision capacity of the Duero river, thus favoring the preservation of large relicts of the endorheic morphology in the upstream part of this basin.

Author contributions
AV undertook morphometric modeling and interpretation, and wrote the paper. SB and FM contributed to the interpretation and the writing.

Competing interests.
The authors declare that they have no conflict of interest.

Acknowledgements.
This study was funded by the OROGEN Project, a TOTAL-BRGM-CNRS consortium. We
thank two reviewers and associated Editor Veerle Vanacker for very useful and constructive
comments that greatly helped us to clarify and improve this manuscript.

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

Figure captions:

Figure 1: A) Topographic map of the Duero and Ebro basins and surrounding belts. B) Averaged topographic section throughout the Duero and Ebro basins showing important incision contrast between the two basins. The Duero basin recorded low incision, especially in its upper part, whereas the Ebro basin is highly excavated.

Figure 2: Simplified geological map of the study area.

Figure 3: Topographic map of the study area with all the rivers considered in this study. The red lines represent drainage divides between main hydrographic basins.

Figure 4: Zoom in the geological map of the Iberian Range showing the location of the Jalon river tributaries. The river long profiles of these streams and the location of knickpoints are shown to the left.

Figure 5: A) Zoom in the geological map of the Bureba sector. B) Zoom in the Homino river (Ebro tributary) capturing the upper reach of the Jordan river (Duero tributary). C) Schematic representation of this capture using river long profiles and map orientation, showing the associated knickpoint and wind gap.

Figure 6: Mean annual precipitation map for the study area (data from Hijmans et al., 2005).

Figure 7: A) 3D view of the DEM of the Bureba sector showing important contrast of incision between the Ebro and Duero basins across their divide (red dashed line) and river capture evidence (elbows of capture, knickpoints and wind gaps). B) Google Earth image around the locality of Hontomin where the Homino river is capturing the upper reach of the Jordan river. C) and D) Wind gaps cut into the Bureba anticline (see location on Fig. 7A). Pictures have been taken from the north of this structure toward the south. E) Possible three steps evolution of the southwestward divide retreat through multiple river captures witnessed in the area.

Figure 8: River long profiles for all the streams described in the Bureba area showing evidence of river capture. Colors are given to rivers that are linked in these capture processes.

Figure 9: Topographic map showing the location of all the knickpoints and low relief surfaces that may be associated to river capture. The black dashed line represents a possible paleodrainage divide between the Ebro and Duero basins. The area between this dashed line and the present-day location of the divide in red may have belonged the Duero basin before being captured by the Ebro basin.

Figure 10: Duero river long profile (black line) and difference in the specific stream power of the river (grey) calculated by considering the paleo and present-day position of its divide. Positive values suggest a significant diminution of the incision capacity of the Duero river, particularly along the knickzone of its longitudinal profile. Details on calculation are available in the Supplement (Section S1).

Figure 11: Topographic map with $\chi$ values calculated on different opposite streams in the vicinity of the Ebro/Duero drainage divide. This map shows significant contrasting values between the Ebro and Duero drainage networks.

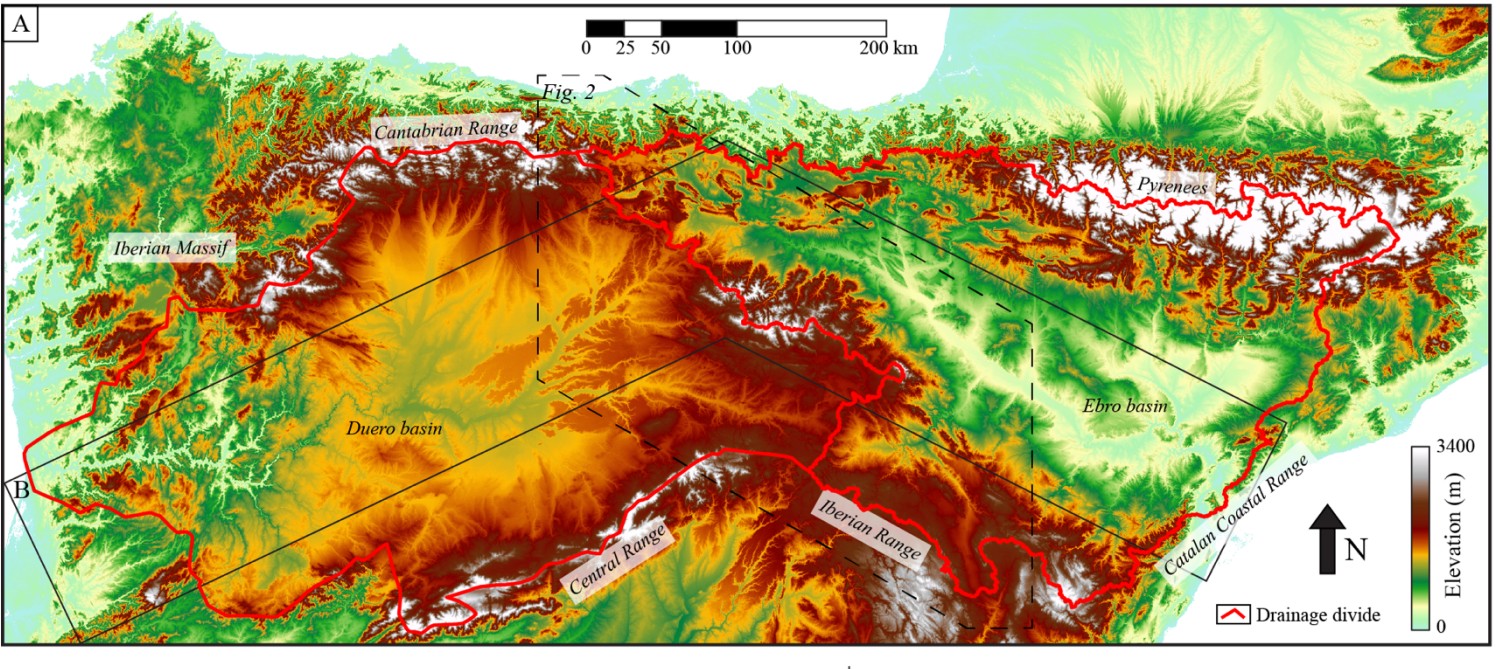

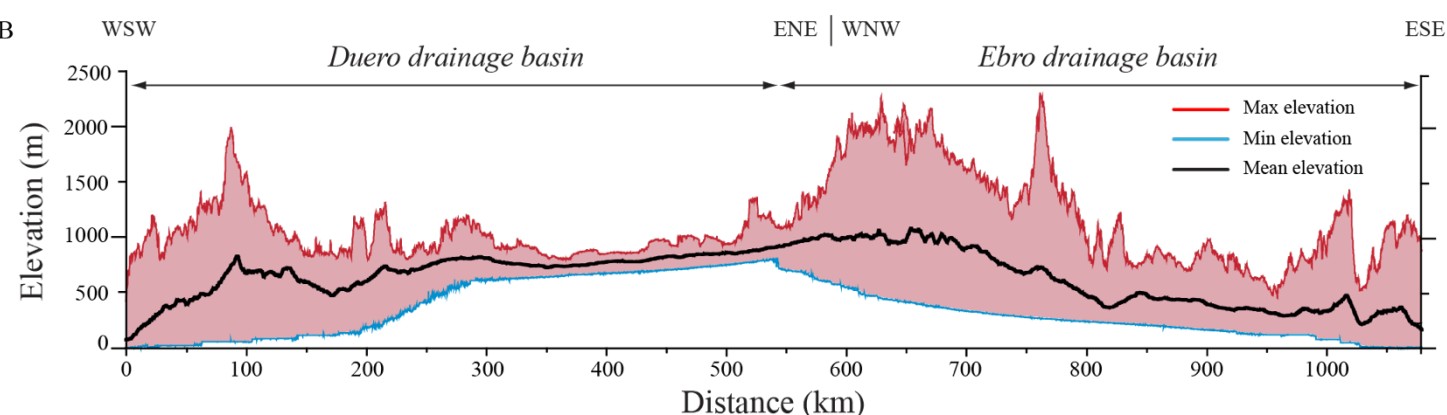

Figure 1

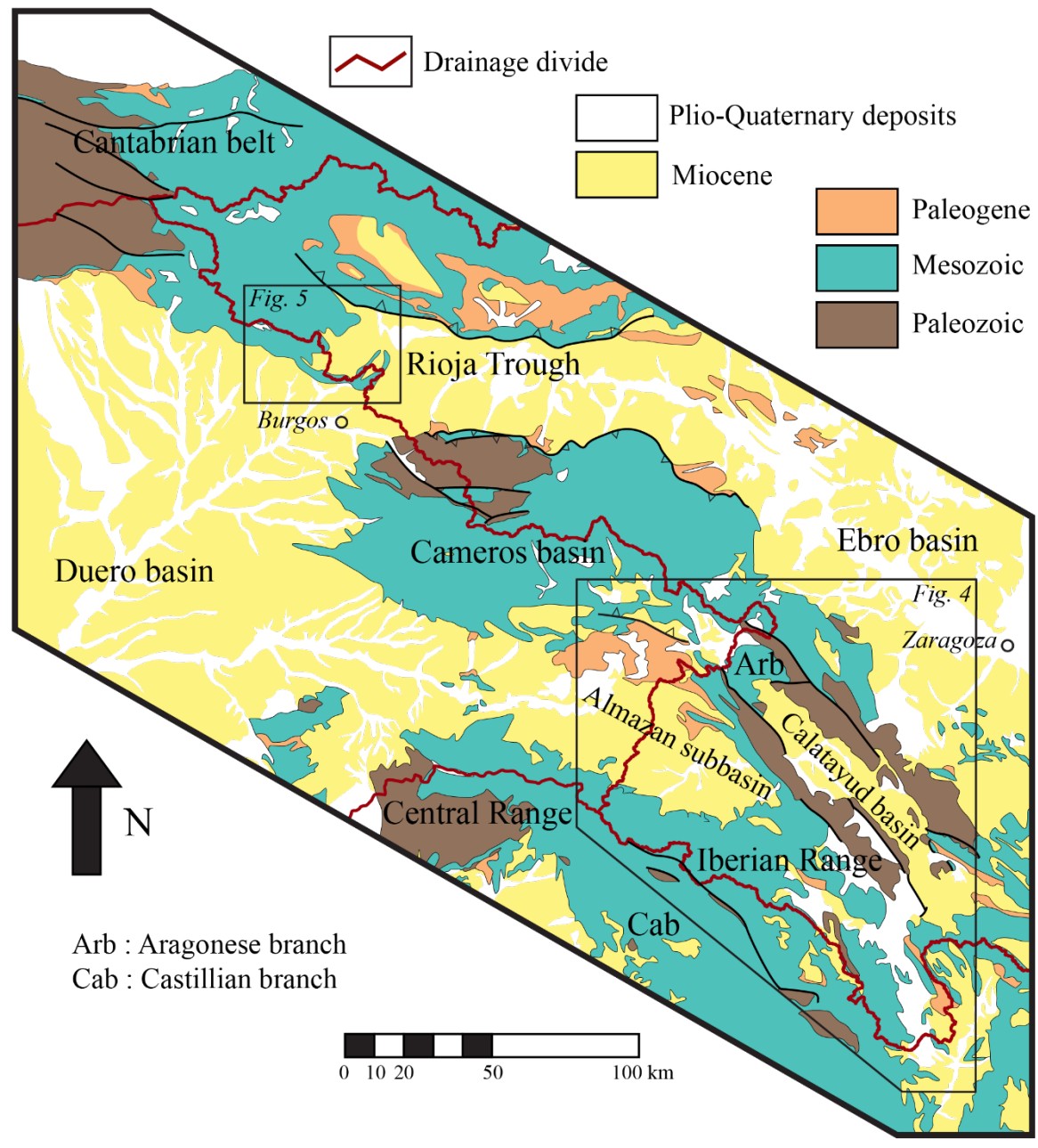

Figure 2

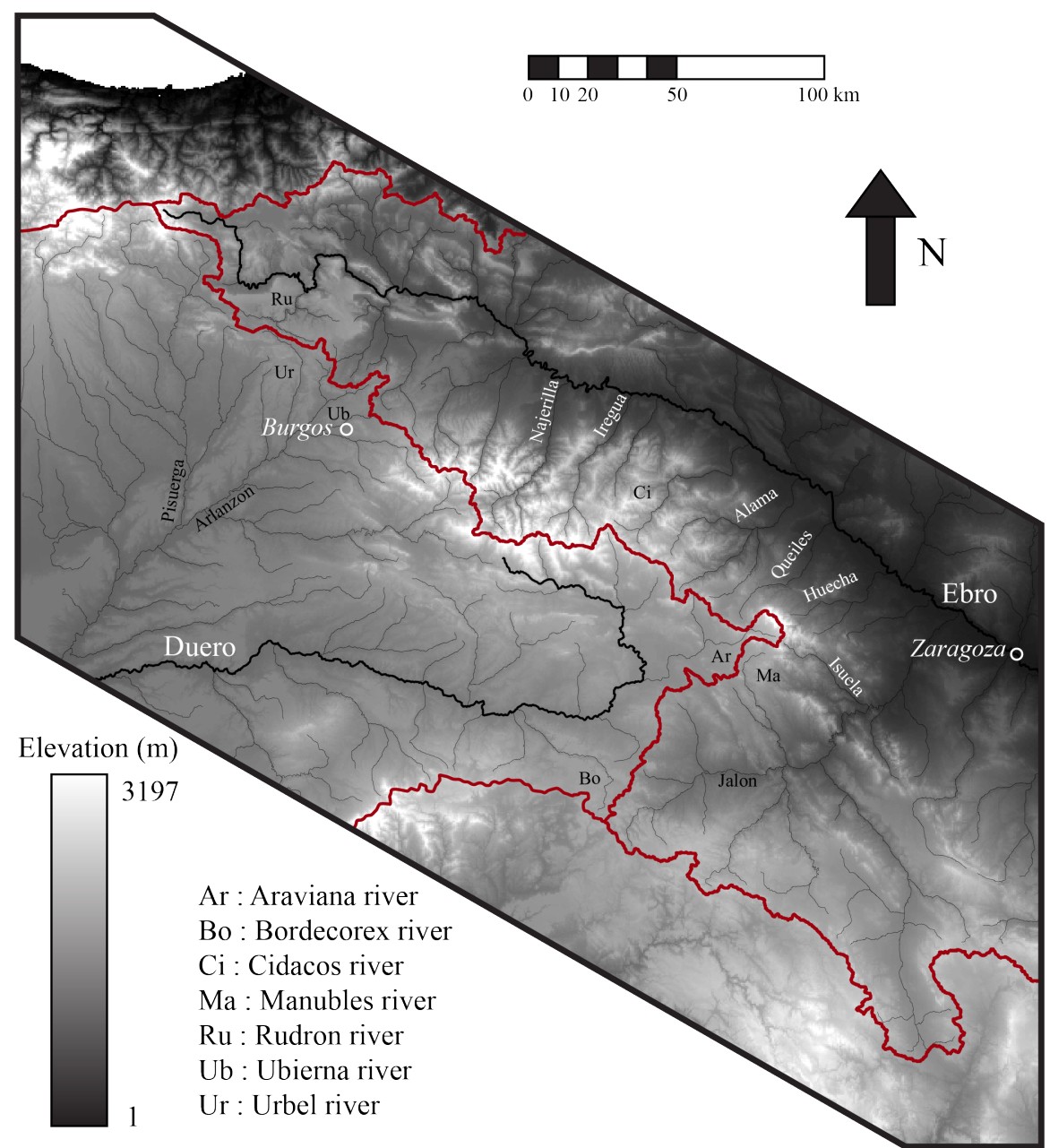

Elevation (m)

3197

Ar : Araviana river
Bo : Bordecorex river
Ci : Cidacos river
Ma : Manubles river
Ru : Rudron river
Ub : Ubierna river
Ur : Urbel river

Figure 3

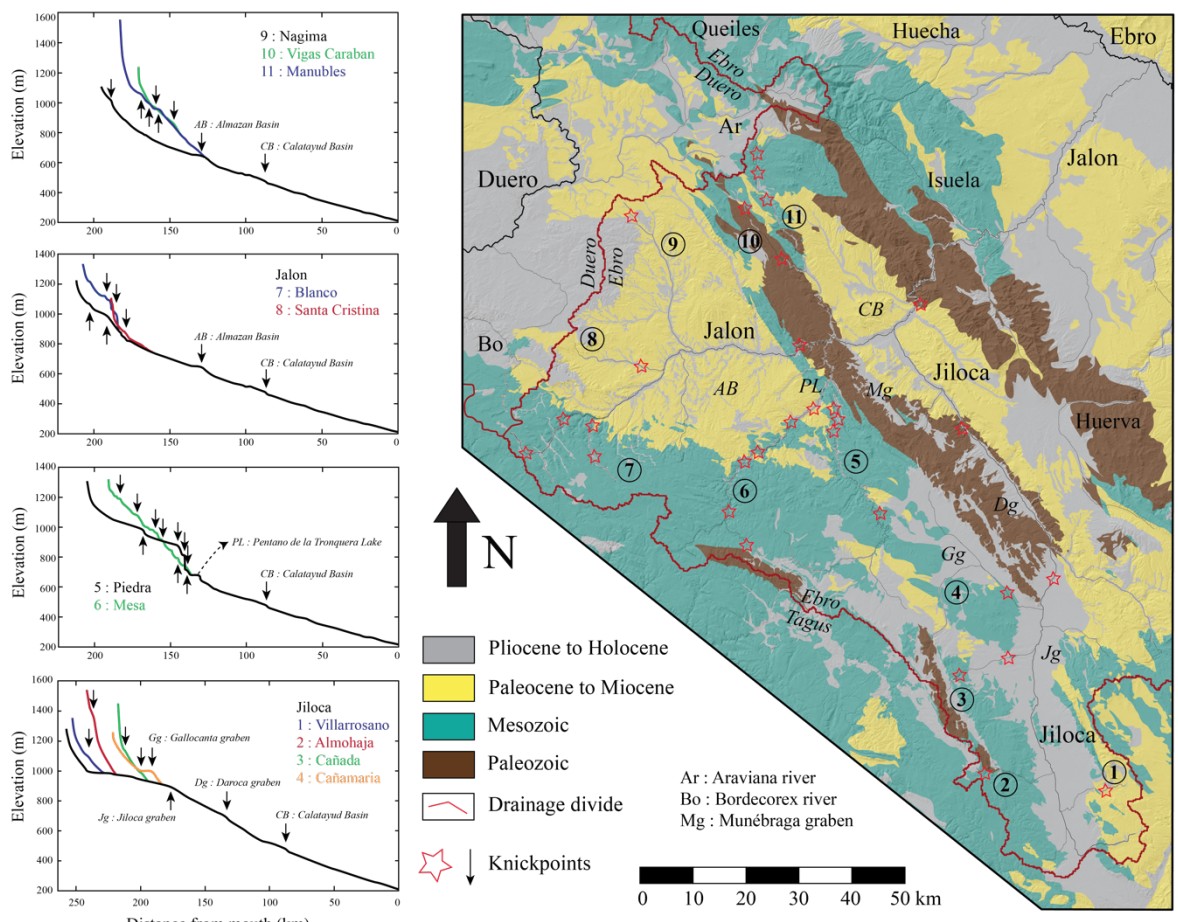

Figure 4

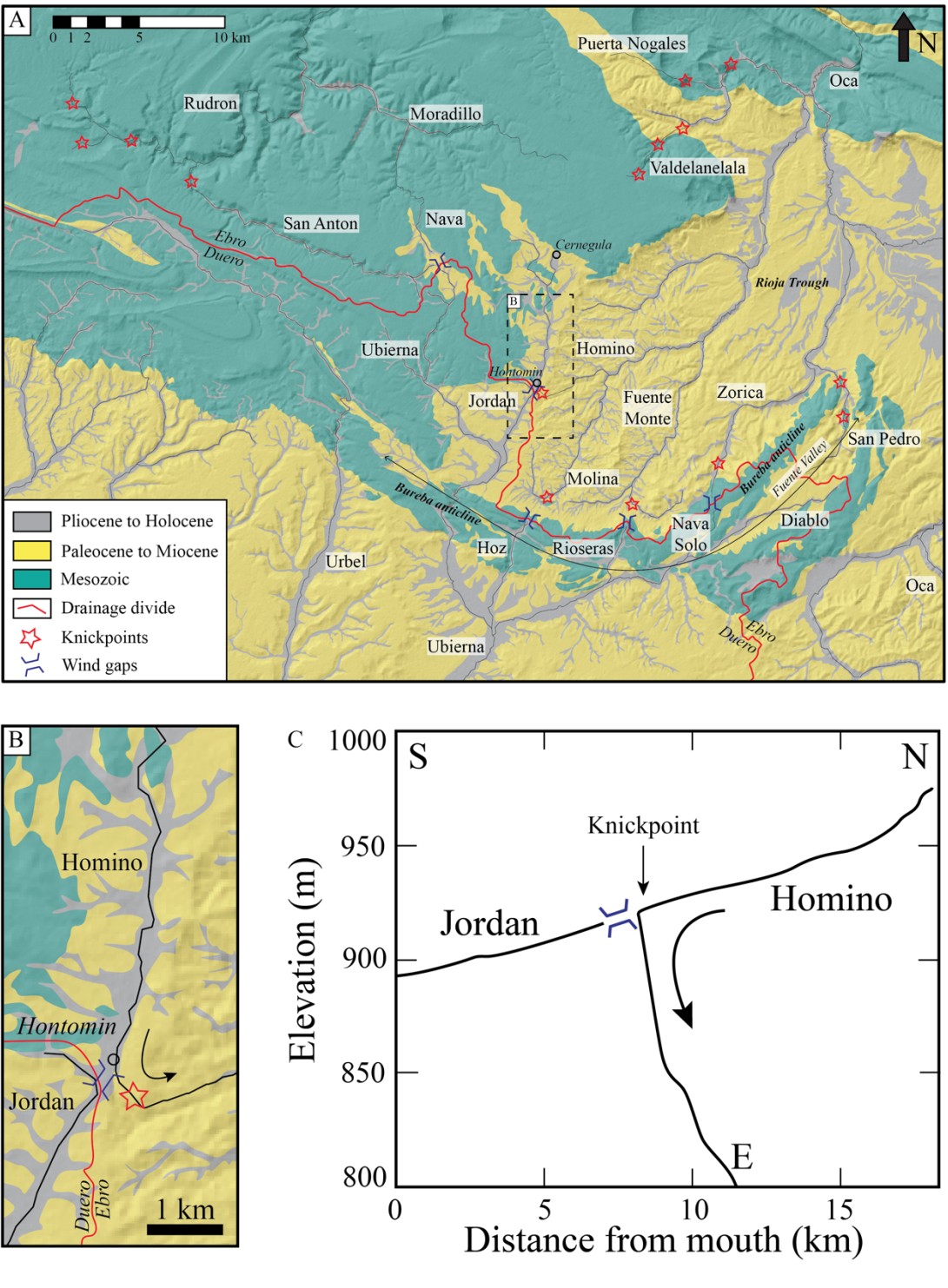

Figure 5

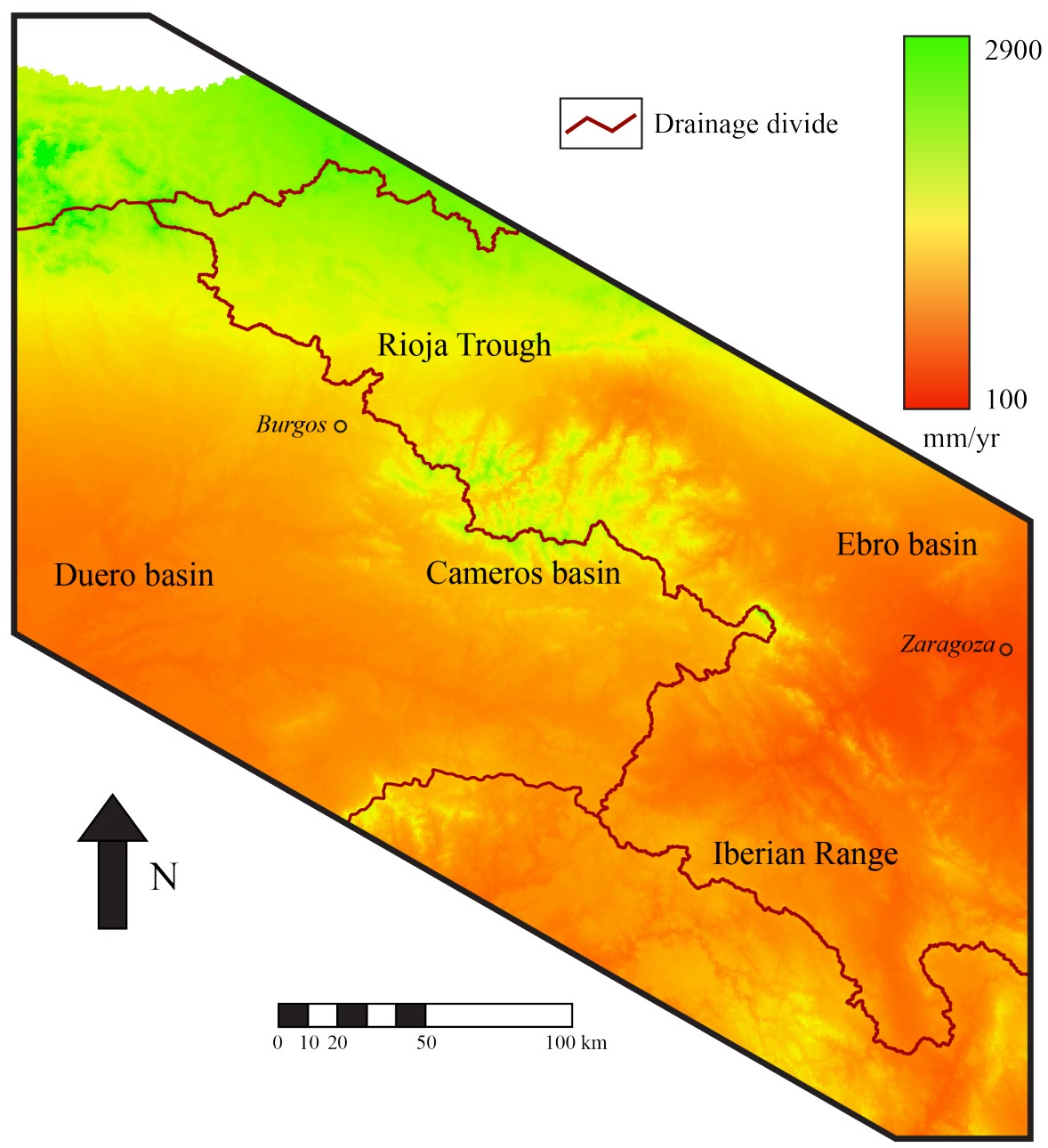

Figure 6

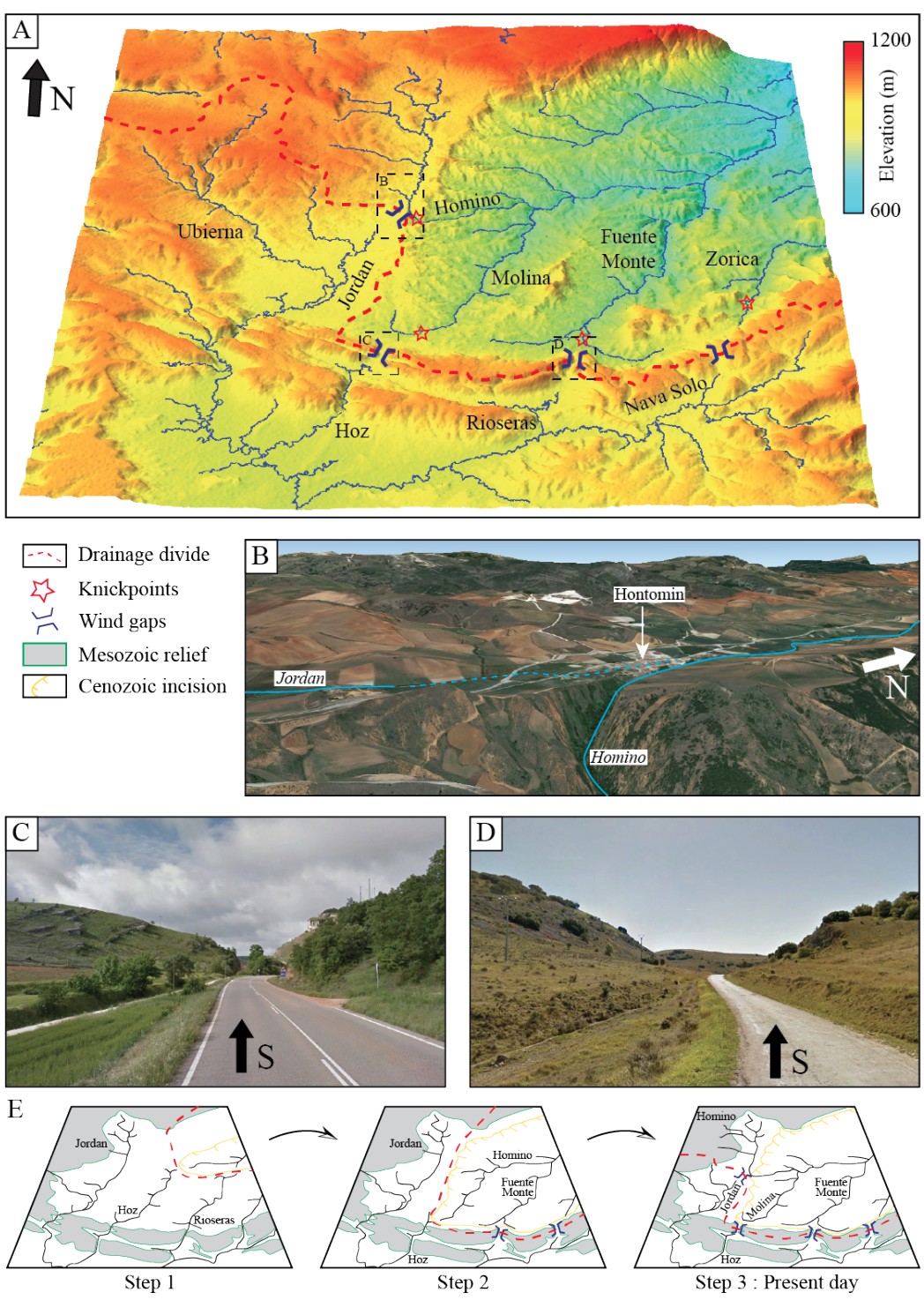

Figure 7

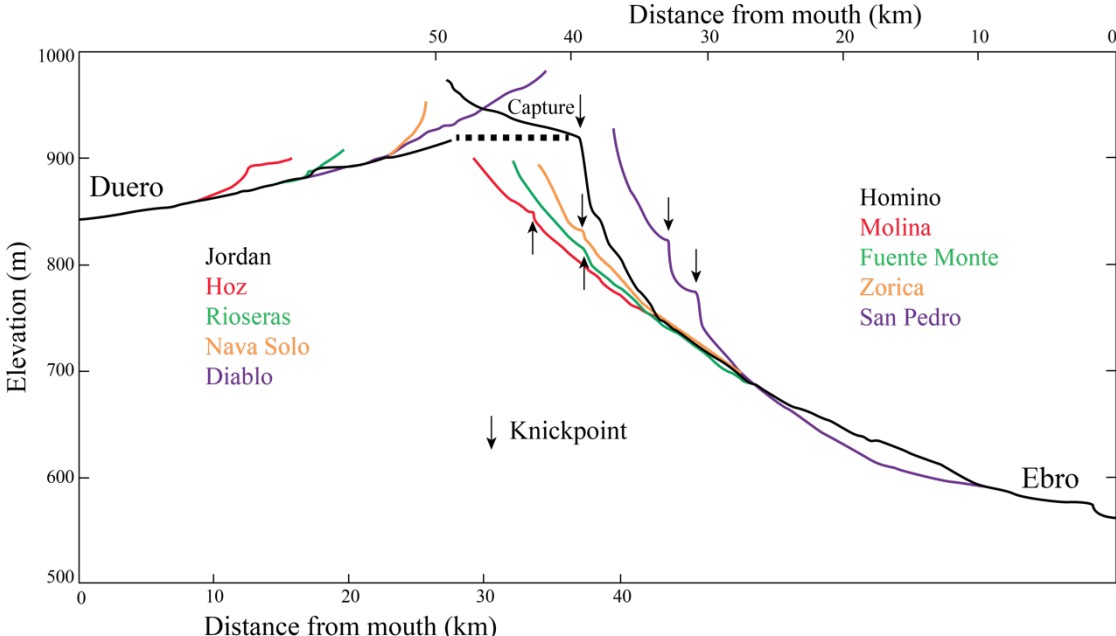

Figure 8

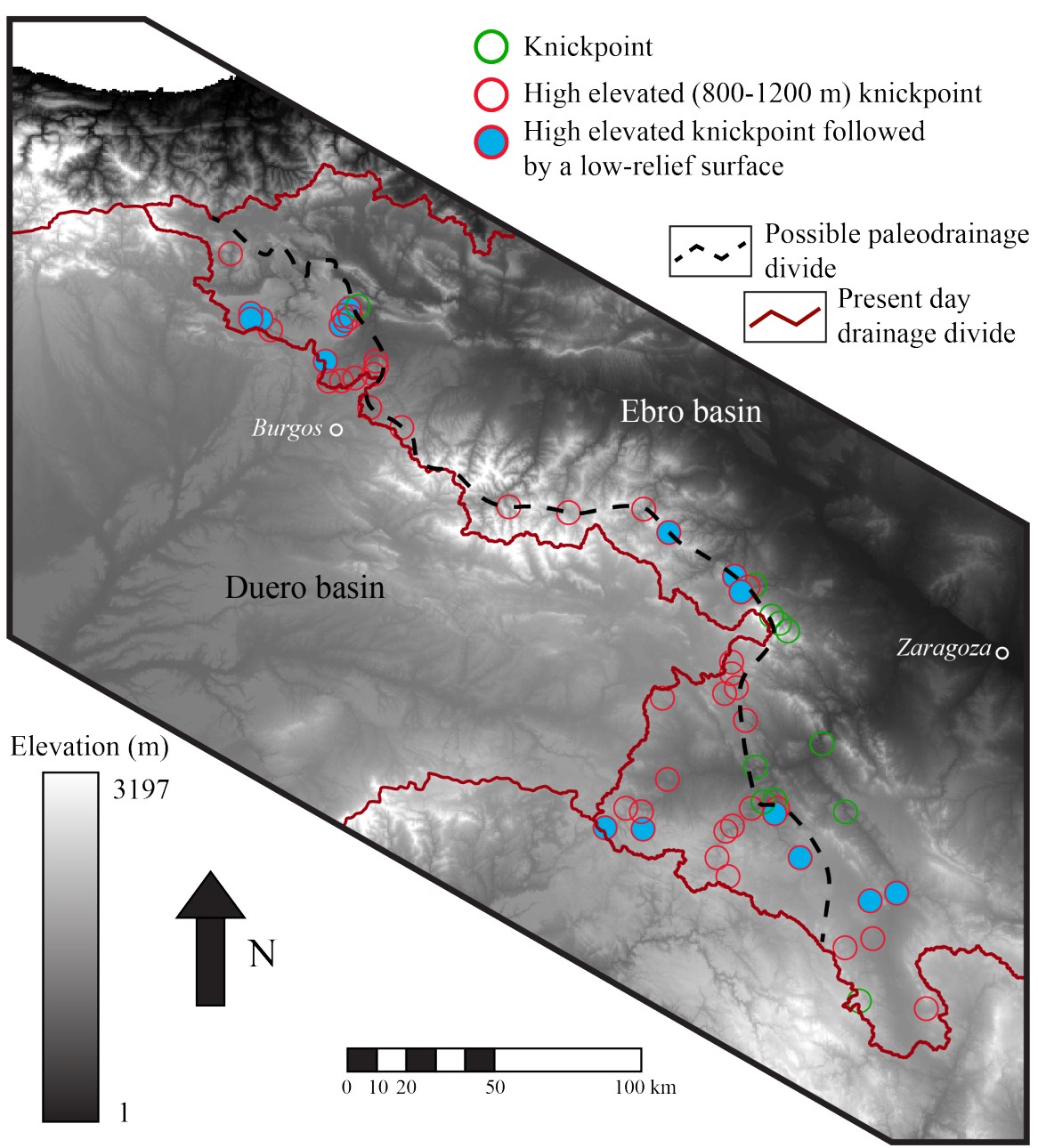

Figure 9

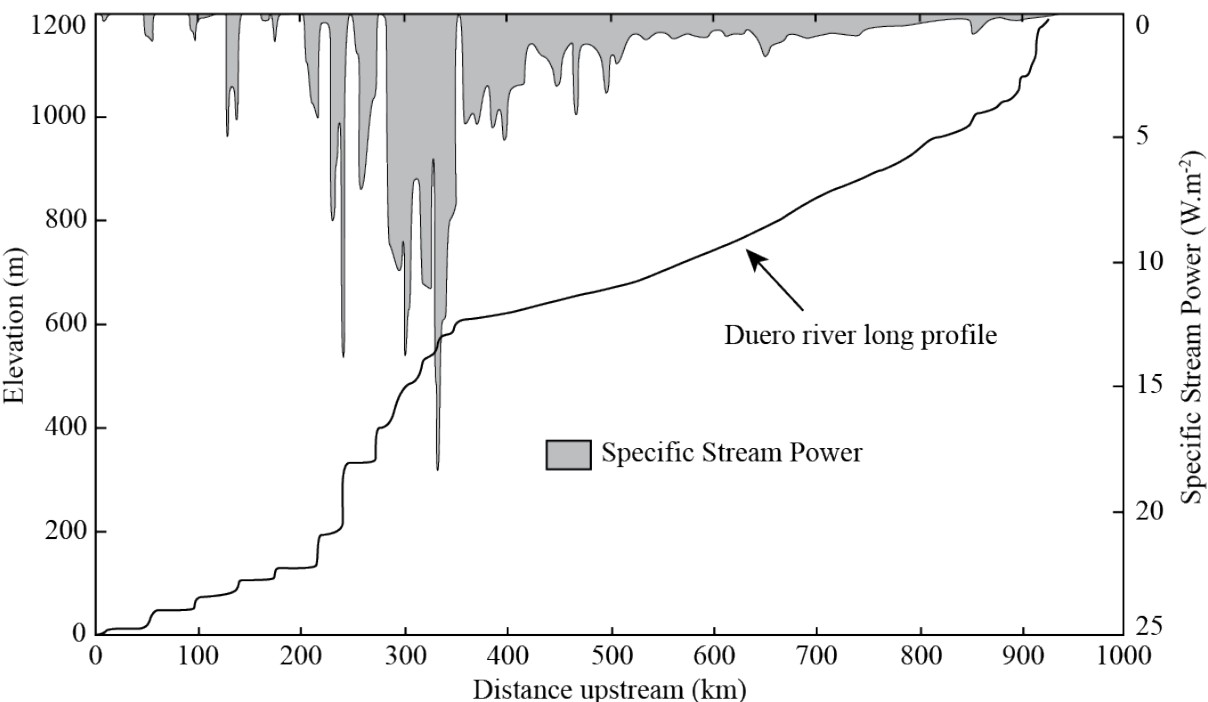

Figure 10

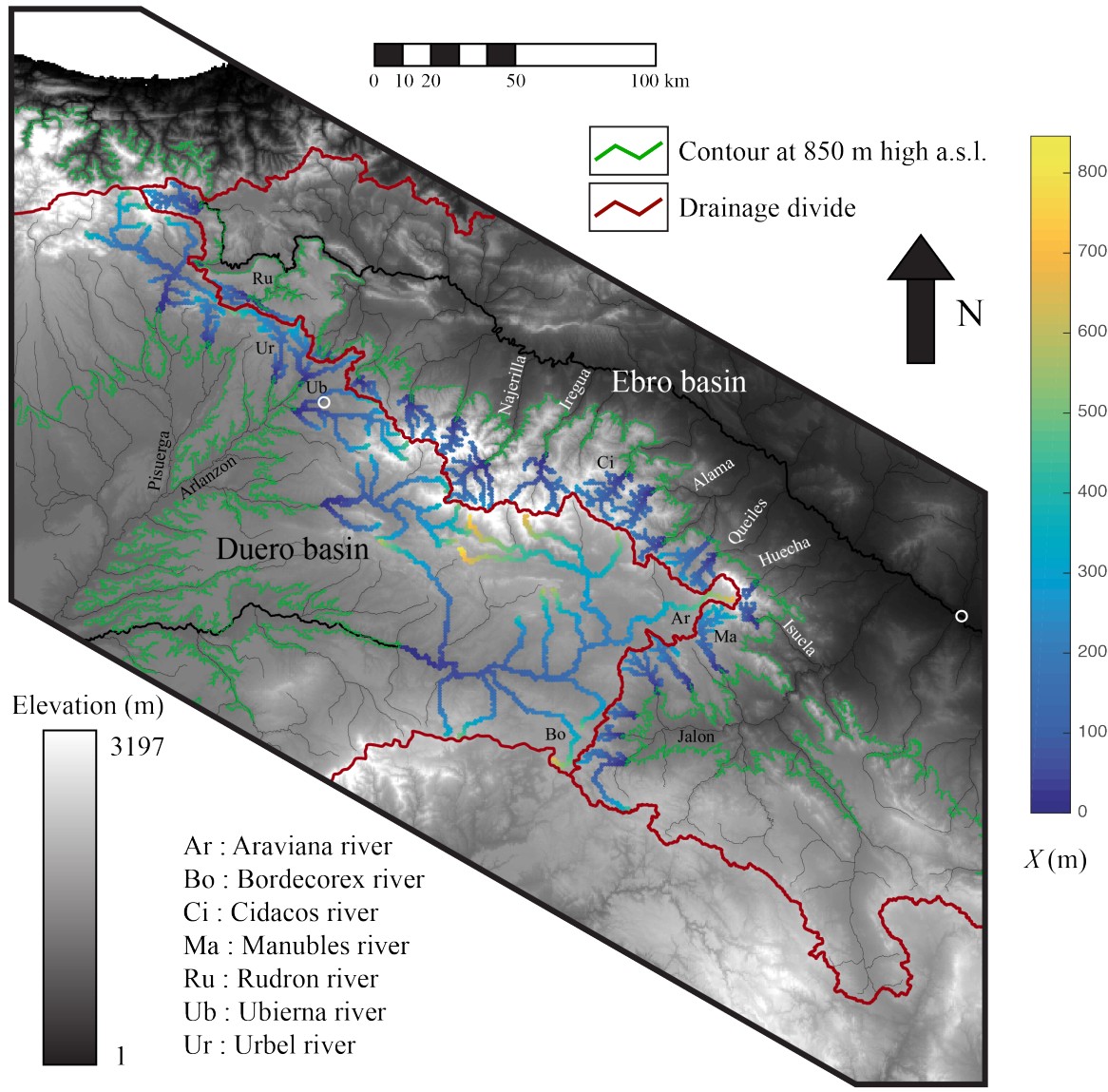

Figure 11

**Drainage reorganization and divide migration induced by the excavation of the Ebro**
**basin (NE Spain)**
Arnaud Vacherat [1], Stéphane Bonnet [1], Frédéric Mouthereau [1]
[1]Géosciences Environnement Toulouse (GET), Université de Toulouse, CNRS, IRD, UPS,
(Toulouse), France
*Correspondance to*: Stéphane Bonnet (stephane.bonnet@get.omp.eu)
**Abstract**
Intracontinental endorheic basins are key elements of source-to-sink systems as they preserve
sediments eroded from the surrounding catchments. Drainage reorganization in such a basin in
response to changing boundary conditions has strong implications on the sediment routing
system and on landscape evolution. The Ebro and Duero basins represent two foreland basins,
which developed in response to the growth of surrounding compressional orogens, the Pyrenees
and the Cantabrian mountains to the north, the Iberian Ranges to the south, and the Catalan
Coastal Range to the east. They were once connected as endorheic basins in the early Oligocene.
By the end of the Miocene, new post-orogenic conditions led to the current setting in which the
Ebro and Duero basins are flowing in opposite directions, towards the Mediterranean Sea and
the Atlantic Ocean. Although these two hydrographic basins recorded a similar history, they are
characterized by very different morphologic features. The Ebro basin is highly excavated,
whereas relicts of the endorheic stage are very well preserved in the Duero basin. The
contrasting morphological preservation of the endorheic stage represents an ideal natural
laboratory to study the drivers (internal / external) of post-orogenic drainage divide mobility,
drainage network and landscape evolution. To that aim, we use field and map observations and
we apply the χ-analysis of river profiles along the divide between the Ebro and Duero drainage
basins. We show here that the contrasting excavation of the Ebro and Duero basins drives a
reorganization of their drainage network through a series of captures, which resulted in the
southwestward migration of their main drainage divide. Fluvial captures have strong impact on
drainage areas, fluxes, and so on their respective incision capacity. We conclude that drainage
reorganization driven by the capture of the Duero rivers by the Ebro drainage system explains
the first-order preservation of endorheic stage remnants in the Duero basin, due to drainage area
loss, independently from tectonics and climate.
**1. Introduction**
Landscapes subjected to contrasted erosion rates between adjacent drainage basins show a
migration of their drainage divide toward the area of lower erosion rates (Bonnet, 2009; Willett
et al., 2014). This is the case for mountain ranges characterized by gradients in precipitation
rates due to orography, once landscapes are in a transient state and are not adjusted to
precipitation differences (Bonnet, 2009). It also occurs when drainage reorganized in response
to capture (Yanites et al., 2013; Willett et al., 2014). River capture actually drives a drop in the
spatial position location of drainage divide (Prince et al. 2011) but also produces a wave of
erosion in the captured reach (Yanites et al., 2013) that may impact divide position. Historically,
migration of divides has been inferred by changes in the provenance of sediments stored in
sedimentary basins (*e.g.* Kuhlemann et al., 2001). It is however a process that is generally very
difficult to document in erosional landscapes. Recent developments have provided models and
analytical approaches to identify divide migration in the landscape (Bonnet, 2009; Castelltort
et al., 2012; Willett et al., 2014; Whipple et al., 2017). Among them the recently-developed χ-
analysis of longitudinal profiles of rivers (Perron and Royden, 2012) is based on the recognition
of disequilibrium along river profiles, disequilibrium being defined by the departure from an
ideal equilibrium shape. The application of this method to both natural and numerically-
simulated landscapes, has allowed to demonstrate contrasts in the equilibrium state of rivers
across divide and then to infer their migration (Willett et al., 2014). The applicability of this
method is however limited to settings where the response time of rivers is larger compared to
the rate of divide migration, so they can actually show disequilibrium in their longitudinal
profiles (Whipple et al., 2017).
The Ebro and Duero drainage basins in the Northern Iberian Peninsula show geological and
geomorphological evidence of very contrasted erosional histories during the Neogene. They
initially recorded a long endorheic stage from the Early Oligocene to the Late Miocene (Riba
et al., 1983; Garcia-Castellanos et al., 2003). Since then, both basins opened toward the Atlantic
Ocean (Duero) or the Mediterranean Sea (Ebro). The Ebro basin's opening is reflected in the
landscape by evidence of river incision (Garcia-Castellanos et al., 2003), whereas the Duero
Basin does not show significant incision in its upstream part as a large relict of its endorheic

morphology is preserved (Antón et al., 2012). The Duero river long profile actually shows a pronounced knickpoint (knickzone) defining an upstream domain of high mean elevation (~800 m) and low relief where the sediments deposited during the endorheic stage are relatively well preserved. Then, these two adjacent basins are characterized by differences in incision and in the preservation of their endorheic stages. They thus represent an ideal natural laboratory to evaluate divide migration in response to differential post-orogenic incision. Following a presentation of the geological context, we first compile evidence of fluvial captures along the Ebro-Duero divide, based on previous studies and our own investigations, and we map the location of knickpoints and relict portions of the drainage network. We use all these observations to reconstruct a paleo-divide position and to estimate the impact of divide migration in terms of drainage area and stream power. We complement this dataset by providing a map of χ across divide (Willett et al., 2014) to highlight potential disequilibrium state between rivers of the Ebro and Duero catchments.

**2. Geological setting**

2.1 The Ebro and Duero basins

The Ebro and Duero basins represent two hydrographic basins located in the northern part of the Iberian Peninsula (Fig. 1). The bedrock of the Ebro and Duero drainage basins mainly consists of Cenozoic deposits, and Mesozoic and Paleozoic rocks in their headwaters (Fig. 2). They formed once a unique foreland basin during the Cenozoic controlled by the flexural loading by the surrounding mountain belts: the Pyrenees and the Cantabrian mountains to the north (Pulgar et al., 1999), the Iberian and Central Ranges to the south (Guimerà et al., 2004; De Vicente et al., 2007), and the Catalan Coastal Range (CCR) to the east (López-Blanco et al., 2000 ; Salas et al., 2001), during collision between Iberia and Europe since the Late Cretaceous.

From the Late Cretaceous, the Ebro and Duero basins were essentially filled by clastic deposits, and opened toward the Atlantic Ocean in the Bay of Biscay (Alonso-Zarza et al., 2002). During the Late Eocene – Early Oligocene, the uplift in the Western Pyrenees (Puigdefàbregas et al., 1992) led to the closure of the Ebro and Duero basins as attested by the Ebro basin continentalization dated at ~36 Ma (Costa et al., 2010). The center of these two basins became long-lived lakes filled with lacustrine, sandy, and evaporitic deposits from the Oligocene to the Miocene (Riba et al., 1983; Alonso-Zarza et al., 2002; Pérez-Rivarés et al., 2002, 2004; Garcia-

Castellanos et al, 2003; Garcia-Castellanos, 2006; Larrasoaña et al., 2006; Vázquez-Urbez et
al., 2013). The opening of the Ebro basin through the Catalan Coastal Range toward the
Mediterranean Sea occurred during the Late Miocene, leading to kilometer-scale excavation
throughout the basin (Fillon and Van der Beek, 2012; Fillon et al., 2013; Garcia-Castellanos
and Larrasoaña, 2015). The exact timing and and processes driving the opening, as well as the
role of the Messinian Salinity Crisis, have long been debated (Coney et al., 1996 (post-
Messinian); Garcia-Castellanos et al., 2003 (13-8.5 Ma); Babault et al., 2006 (post-Messinian);
Urgeles et al., 2010; Cameselle et al. (2014) (Serravallian-Tortonian); Garcia-Castellanos and
Larrasoaña, 2015 (12-7.5 Ma)). In contrast with the Ebro basin, incision in the upper Duero
basin appears much less significant. The Duero basin is characterized by a low relief topography
(Fig. 1) in its upstream part, at 700-800 m above sea level to the west, and at 1000-1100 m a.s.l.
to the north, northeast, and to the east in the Almazan subbasin, close to the divide with the
Ebro basin. The connection of the Duero River with the Atlantic Ocean occurred from the Late
Miocene-Early Pliocene to the Late Pliocene-Early Pleistocene (Martín-Serrano, 1991). The
current Ebro and Duero drainage networks are separated by a divide running from the
Cantabrian belt to the NW, toward the SE in the Iberian Range (Figs. 1, 2, 3). In the following,
we review the geological evolution of the different domains that constitute this drainage divide
between the Ebro and Duero drainage basins.
2.2 The Iberian Range
The Iberian Range (Figs. 2, 4) is a double vergent fold-and-thrust belt resulting from Late
Cretaceous inversion of Late Jurassic-Early Cretaceous rift basins during Iberia – Europe
convergence (Salas et al., 2001; Guimerà et al., 2004; Martín-Chivelet et al., 2002). It is divided
into two NW-SE directed branches, the Aragonese and the Castillian branches, separated by the
Tertiary Almazan subbasin (Bond, 1996). The Almazan subbasin is connected to the Duero
basin since the Early Miocene (Alonso-Zarza et al., 2002).
The Iberian Range is essentially made of marine carbonates and continental clastic sediments
ranging from Late Permian to Albian, overlying a Hercynian basement. The Cameros subbasin
to the NW represents a late Jurassic-Early Cretaceous trough almost exclusively filled by
continental siliciclastic deposits (Martín-Chivelet et al., 2002 and references therein; Del Rio
et al., 2009). Shortening in the Iberian Range occurred from the Late Cretaceous to the Early
Miocene, along inherited Hercynian NW-SE structures (Gutiérrez-Elorza and Gracia, 1997;
Guimerà et al., 2004; Gutiérrez-Elorza et al., 2002). The opening of the Calatayud basin in the
Aragonese branch occurred during the Early Miocene in response to right-lateral transpression
on the southern margin of the Iberian Range (Daroca area) (Colomer and Santanach, 1988). It
is followed during the Pliocene and the Pleistocene, by pulses of extension reactivating faults
in the Calatayud basin, and the formation of grabens such as the Daroca, Munébrega,
Gallocanta, and Jiloca grabens (Fig. 4; Colomer and Santanach, 1988; Gutiérrez-Elorza et al.,
2002; Capote et al., 2002). This is also outlined by the occurrence of Late Pliocene to Early
Pleistocene breccias and glacis levels in the Daroca and Jiloca grabens (Gracia, 1992, 1993a;
Gracia and Cuchi, 1993; Gutiérrez-Santolalla et al., 1996). These Neogene troughs are filled by
continental deposits and pediments, up to the Quaternary (Fig. 4). The Neogene tectonic pulses
in the Iberian are interrupted by periods of quiescence during which erosion surfaces developed
(Gutiérrez-Elorza and Gracia, 1997).
Deformation and uplift of the Iberian Range and Cameros basin resulted in the development of
a new drainage divide between the Duero and Ebro basins and in the isolation of the Almazan
subbasin (Alonso-Zarza et al., 2002). In contrast, the connection between the Duero the Ebro
basins has not been affected by significant deformation and uplift in the proto-Rioja trough
(Mikes, 2010).

2.3 The Rioja trough and Bureba high

The Rioja trough (Figs. 2, 5) recorded important subsidence, especially during the Cenozoic (>
5 km), related to compression and thrusting on its borders (Jurado and Riba, 1996). As thrusting
initiated in the Pyrenean-Cantabrian belt and in the Iberian Range and Cameros basin, the Rioja
trough became domain of important synorogenic sediment transfer between the Ebro and Duero
basins. During the Paleocene, the Rioja trough was a marine depositional environment. With
the increase of sediment fluxes that originated from the exhumation of surrounding mountain
bets, sedimentation became essentially continental in the Eocene. Thrusting continued during
the Oligocene resulting in the formation of an anticline connecting the Cantabrian domain and
the Cameros inverted basin. This morphologic high (the Bureba anticline, Fig. 5) located in the
center of the area is supposed to have triggered the disconnection between the Duero and Ebro
basins (Mikes, 2010), as suggested by the repartition of alluvial fans on both sides of this
structure (Muñoz-Jiménez and Casas-Sainz, 1997; Villena et al., 1996). During the Miocene,
deformation ceased as evidenced by the deposition of undeformed middle Miocene to Holocene
strata. The Bureba anticline is cored by Albian strata and topped by Santonian limestones and
Oligocene conglomerates controlling the location of the current main drainage divide between
the Ebro and Duero river networks (Fig. 5).

2.4 Climate evolution

Climate exerts a major control on valley incision, sediment discharge, and on the evolution of
drainage networks (Willet, 1999; Garcia-Castellanos, 2006; Bonnet, 2009; Whipple, 2009;
Whitfield and Harvey, 2012; Stange et al., 2014). The mean annual precipitation map for the
North Iberian Peninsula (Hijmans et al., 2005) shows a similar pattern for both the Ebro and
Duero basins as they record very low precipitation, associated with global subarid conditions,
with the exception of the Cameros basin that record a slightly higher precipitation rate (Fig. 6).
There is a strong contrast to the north, toward the Mediterranean Sea and the most elevated
areas in the Cantabrian and Pyrenean belts, where precipitation drastically increases.
The paleoclimatic evolution from the Late Cretaceous to the Neogene is linked both with the
effects of surrounding mountains uplift, and with the latitudinal variation drift of Iberia from
30°N in the Cretaceous to ~40°N during Late Neogene times. The hot-humid tropical climate
of the Late Cretaceous became drier and arid from the Paleocene to the Middle Miocene (López-
Martínez et al., 1986), favouring the development of endorheic lakes (Garcia-Castellanos,
2006). During the Middle-Late Miocene and Early Pliocene, the northern Iberia recorded more
humid and seasonal conditions (Calvo et al., 1993; Alonso-Zarza and Calvo, 2000) with
alternations of cold-wet and hot-dry periods (Bessais and Cravatte, 1988; Rivas-Carballo et al.,
1994; Jiménez-Moreno et al., 2010). More humid and colder conditions took place in the Late
Pliocene, characterized by dry glacial periods and humid interglacials (Suc and Popescu, 2005;
Jiménez-Moreno et al., 2013). Climatic contrasts increased, triggering intense glaciers
fluctuations in the surrounding mountain ranges during the Lower-Middle Pleistocene transition
(1.4-0.8 Ma) (Moreno et al., 2012; Duval et al., 2015; Sancho et al., 2016), and throughout the
Late Pleistocene period, which record glacial / interglacial oscillations, as evidenced by pollen
identification (Suc and Popescu, 2005; Jiménez-Moreno et al., 2010, 2013; Barrón et al., 2016;
García-Ruiz et al., 2016) and speleothem studies (Moreno et al., 2013; Bartolomé et al., 2015).
Glaciers are considered as very efficient erosion tool in continental environment. They are
likely to influence drainage divide migration (Brocklehurst and Whipple, 2002). There is large
evidence of glaciers development especially for the Late Pleistocene in the Pyrenees (Delmas
et al., 2009; Nivière et al., 2016; García-Ruiz et al., 2016), in the Cantabrian belt (Serrano et
al., 2013, 2016; García-Ruiz et al., 2016), and in the Central Range (Palacios et al., 2011, 2012;
García-Ruiz et al., 2016). However, although numerous moraines have been mapped throughout
the Iberian Range (Ortigosa, 1994; García-Ruiz et al., 1998; Pellicer and Echeverría, 2004),
there is no evidence of U-shaped valleys and because of the lack of very high elevated massifs
(>2500 m), the occurrence of active ice tongues are considered as limited, if not precluded
(García-Ruiz et al., 2016).

**3. Evidence of divide mobility between the Duero and Ebro catchments**

The easternmost part of the Duero river is opposed to the Ebro tributaries that are the Jalon,
Huecha, Queiles, Alama, Cidacos, Iregua, and Najerilla rivers, whereas the Arlanzon and
Pisuerga rivers (Duero tributaries) are opposed to the Najerilla, Tiron, Oca, and Rudron rivers,
and to the westernmost part of the Ebro river (Fig. 3). The northeastern part of the Duero basin
(the easternmost Duero river, the Arlanzon and Pisuerga rivers) mainly consists of broad flat
valleys characterized by low incision depth and low-gradient streams with concave longitudinal
profiles (Antón et al., 2012, 2014). By contrast, the western part of the Ebro basin is
characterized by more incised valleys, especially in the Cantabrian and in the Cameros – Iberian
Range domains, with more complex longitudinal profiles (knickpoints, remnants of high
elevated surfaces). Previous studies (Gutiérrez-Santolalla et al., 1996; Pineda, 1997; Mikes,
2010) already shown that the Jalon and Homino rivers, which belong to the Ebro basin, have
recently captured parts of the Duero basin in the Iberian Range and in the Rioja trough,
respectively. Such evolution has been recorded by the occurrence of geomorphological markers
as wind gaps and elbows of captures, as well as by the presence of knickpoints and/or remnants
of high elevated surfaces in river long profiles. To highlight this dynamic evolution, we
performed a morphometric analysis of rivers all around the divide separating the Ebro basin
from the Duero basin, with particular attention given to the Aragonese branch of the Iberian
Range (Fig. 4) and to the Rioja Trough (Fig. 5), where captures have already been described.
The studied basins were digitally mapped using high-resolution (~30 meters) digital elevation
models (DEMs) from SRTM 1 Arc-Second Global elevation data available at the U.S.
Geological Survey (www.usgs.gov). The different DEMs were assembled using the ENVI
software. We also used 1:50,000 geological maps from the Instituto Géologico y Minero de
España (www.igme.es). We used the TopoToolbox, a MATLAB-based software developed by
Schwanghart and Scherler (2014), to extract the river network and longitudinal profiles and the
$\chi$-analysis Tool developed by Mudd et al. (2014).

3.1 Fluvial captures and related knickpoints in the Iberian Range

Neogene tectonics in the Iberian range controlled the uplift of topographic ranges and the
formation of several basins whose connection with the Ebro or the Duero has occasionally
changed through time. Nowadays, the western part of the Almazan subbasin (Figs. 2, 4) belongs
to the Duero catchment, its eastern part being drained by the Ebro drainage network and
especially by the Jalon river and its tributaries (Fig. 4). Gutiérrez-Santolalla et al. (1996)
proposed that the Jalon river captured this domain after cutting into the Mesozoic and Neogene
strata and the two Paleozoic ridges of the Aragonese branch of the Iberian Range. They used
chronostratigraphic evidence to build a relative chronology of capture events in the Jalon area.
First, the incision of the northern Paleozoic ridge and capture of the Calatayud basin by the
Jalon river is attributed to a post-Messinian age. The Jiloca river, the easternmost main Jalon
tributary, is then thought to capture the Daroca graben area to the east during the Late Pliocene
– Early Pleistocene. This is followed from the Early to Late Pleistocene by the capture of the
Jiloca graben to the southeast and finally by the capture of the Munébraga graben to the
southwest, by the Jalon river (Gutierrez-Santolalla et al., 1996), toward the easternmost part of
the Almazan subbasin.
The Jalon river and tributaries show knickpoints in their longitudinal profiles (Fig. 4), at
locations that are consistent with the events of captures proposed by Gutiérrez-Santolalla et al.
(1996), suggesting that these captures are actually witnessed by knickpoints. The capture of the
Jiloca graben corresponds to a major knickpoint in the Jiloca river profile that appears very
smoothed, and that is followed by an upstream ~50 km long flat domain preserved at ~1000 m
high above sea level. This imparts a convex shape to the Jiloca profile (Fig. 4). Due to the short
period of time between the formation of the Jiloca graben (the earliest glacis deposits are
attributed to the Middle Pliocene) and its capture (Early Pleistocene; Gutierrez-Santolalla et al.,
1996), we suggest this upstream domain was a short-lived endorheic domain that has never
been externally drained before being captured by the Ebro network. In the northwestern part of
the Jiloca graben, the Cañamaria river, a tributary of the Jiloca river, heads to the northwest,
reaching the Gallocanta basin, also considered as a former graben (Gracia, 1993b; Gracia et al.,
1999; Gutiérrez-Elorza et al., 2002). The upstream part of its river long profile is characterized
by a sharper knickpoint at the entrance of the basin, and is followed by a ~15 km long flat
domain (Fig. 4). Similarly to the Jiloca graben, the Gallocanta basin appears to be a short-lived
endorheic domain that has been more recently captured by the Jiloca river network.
According to Gutiérrez-Santolalla et al. (1996), the Jalon river reached the southern Paleozoic
ridge of the Aragonese branch, to the southwest of the Calatayud basin, captured the Munébrega
graben and the Almazan subbasin (also characterized by a pronounced knickpoint) during the
Pleistocene-Holocene, slightly after the capture of the Jiloca graben by the Jiloca river. This is
coherent with morphological analysis of longitudinal profiles, as the major knickpoint related
to the capture of the Jiloca graben appears very smoothed, whereas knickpoints observed in the
west are sharper, suggesting they are younger. However we cannot ruled out some local
influence of the lithology on the shape of these knickpoints.
Finally, the Piedra river (Jalon tributary) long profile shows major sharp knickpoints and two
successive ~30 km long almost flat domains in the Almazan subbasin, at ~900-1000 m above
sea level (Fig. 4). In addition, the upper reach of the river long profiles of the Jalon river, and
of its tributary the Blanco river, are characterized by major sharp knickpoints, and by a ~15 km
long flat domain at ~1000-1100 m above sea level, in the Mesozoic Castillan branch of the
Iberian Range (Fig. 4).

3.2 Fluvial captures and related knickpoints in the Rioja trough area

In the Rioja trough area, the position of the Ebro-Duero divide is partly controlled by the Bureba
anticline. It consists of folded Middle Cretaceous to Early Miocene series, covered by
undeformed Middle Miocene to Holocene deposits (Fig. 5). The anticline is orientated E-W to
the west and NE-SW to the east. The western part of the Rioja trough to the west of the NE-SW
directed branch of the Bureba anticline (Fig. 5), used to be drained toward the Duero basin since
the Oligocene (Pineda, 1997; Mikes, 2010). The westward migration of the divide to its current
location is thought to have occurred in several steps of captures as shown by the occurrence of
remnants of escarpments during the Late Miocene - Pliocene (Mikes, 2010). Once the eastern
branch of the Bureba anticline has been incised, the Ebro tributaries captured the western part
of the Rioja trough, up to the E-W branch of the Bureba anticline to the southwest, from the
Late Miocene to the Pliocene. The western part of the anticline forms a topographic ridge that
is incised by Jordan river (Fig. 5) in a place where the divide between the Ebro and Duero river
networks is located to the north of the ridge. To the East of this location however, the
topographic ridge formed by the Bureba antcline controls the current location of the main
drainage divide (Fig. 5). Here, the ridge exhibits several wind gaps, located on the northward
prolongation of the Hoz, Rioseras, and Nava Solo rivers (Figs. 5, 7). Further east, the Diablo
river does not incise the ridge and its headwater is located in the core of the eastern branch of

the Bureba anticline, the Fuente Valley (Fig. 5). These last streams are tributaries of the Ubierna river, which is a tributary of the Arlanzon river and so, of the Duero river. To the north, the Ebro river system is represented, from west to east, by the Homino river (a tributary of the Oca river) and its four tributaries, the Molina, the Fuente Monte, the Zorica, and the San Pedro rivers (Figs. 5, 7). All these streams are outlined by Late Pleistocene to Holocene alluvial series that are deposited at the bottom of their respective valleys. Valleys from the Duero side appears larger than those from the Ebro side, which are significantly more incised.

The Jordan river's headwater is located north of the ridge formed by the Bureba anticline. We can continuously follow its valley deposits northward along a broadly gentle slope, up to the locality of Coraegula (Fig. 5). However, the current course of the Jordan river is cut ~8 km south, in the vicinity of Hontomin, by the Homino (Ebro) river (Figs. 5B, C, 7). This fluvial capture is characterized by a well-defined and highly incised elbow of capture, already described by Pineda (1997) and Mikes (2010). The longitudinal profile of the Homino river shows a sharp knickpoint located on Hontomin (Fig. 7C). Finally, there is a small wind gap on the divide between the two opposite rivers (Figs. 5, 7).

To the southeast, the headwater of the Hoz river is located to the south of a wind gap cut into the Bureba ridge (Fig. 7C). To the north, in the exact prolongation of the Hoz river, the Molina river shows a bend similar to the elbow of capture previously described for the Homino river (Fig. 7) and there is a minor knickpoint located on this elbow, according to the extracted river long profile. Thus, it is likely that the Molina river used to represent the former upper reach of the Hoz river, in a period when the Ebro-Duero divide was located northward, before being captured by the Ebro network.

To the east, the Rioseras and the Nava Solo rivers have also their headwater located to the South of wind gaps in the Bureba ridge (Fig. 7). Similarly, in their exact prolongations, the Fuente Monte and the Zorica rivers show important elbows of capture with minor knickpoints. They may also represent former upper reaches of Duero streams that have been captured by the Ebro network (Figs. 5, 7, 8).

Further east, the headwater of the Diablo river is located on the depression represented by the core of the eastern branch of the Bureba anticline, the Fuente valley. In its prolongation to the northeast, the San Pedro river incises the northeastern termination of the anticline from the north before entering the valley, leading to a southward retreat of the divide (Fig. 5). Capture is again evidenced by important incision contrast between Ebro and Duero systems, and by sharp knickpoints on the upper reach of the San Pedro river long profile when crossing the Santonian dolomites (Fig. 8). According to this whole set of observations, and in agreement with previous

findings of Pineda (1997) and  Mikes (2010), we propose that the western part of the Rioja trough, in the Bureba area has been recently captured by the Ebro drainage network leading to a sequence of southwestward retreat of the main drainage divide, toward the Duero basin (Fig. 7E).

A similar capture pattern can be observed further west in the continuity of the Bureba anticline (Fig. 5). The San Anton river shows a well-defined elbow of capture accompanied by a smoothed knickpoint (See Fig. S1 in the Supplement) at its junction with the Rudron river (Ebro tributary). The river course is highly incised toward the east, along the northern flank of the WNW – ESE anticline, almost connecting to the upper reach of the Ubierna river. Valley deposits are also observed in the continuity of the Ubierna valley, which former route is suggested by a wind gap (Fig. 5). However, this domain is no longer connected to its network as it is now wandered from the North by the Nava river, a tributary of the Moradillo river, which is a tributary of the Rudron river. This domain clearly records captures leading to divide migration toward the Duero, also in favor of the Ebro basin.

3.3 Past position of the Ebro-Duero divide and implication for stream-power of the Duero River

We used all observations that support divide migration in the Iberian Range and Rioja trough to estimate a paleo-position of the drainage divide between the Duero and Ebro drainage basins (Fig. 9). For this purpose, we considered the location of major knickpoint along the rivers where fluvial captures are defined. Both the Ebro river and several tributaries show high elevated ~10-20 km long flat domains at ~800 – 1200 m a.s.l. and major knickpoints in the upper reach of their long profiles as the Rudron, Queiles, and Alama rivers, as well as the Homino river and its tributaries: the Puerta Nogales and Valdelanelala rivers (Figs. 5, 8; Fig. S1). All these flat domains may not be related to surface uplift as they are not clearly associated with active tectonic features. The Duero basin being characterized by a high mean elevation (~1000 m) and by a very limited incision in the vicinity of the Ebro/Duero drainage divide, a sudden divide migration toward the Duero basin is then expected to isolate such high elevated and relatively preserved surfaces. We suggest these flat domains have been recently captured by Ebro tributaries, and represent remnants of Duero drainage areas, integrated into the Ebro catchment from divide retreat toward the Duero basin. Overall, we consider a paleodrainage divide delimited by these high-elevated knickpoints and flat domains, except for the Jiloca graben area to the southeast, characterized by the occurrence of short-lived endorheic domains (Fig. 9).

Incision in the Ebro basin leads to the capture of new drainage areas, whereas the Duero basin
recorded important loss of its own surface. The present day drainage area of the Cenozoic Duero
basin, upstream of the major knickzone observed to the west in the Iberian Massif is ~63000
km$^2$. We used the paleo-divide position shown in Figure 9 to define a « recent » captured area
that used to belong to the Duero basin. This area represents ~7700 km$^2$, which corresponds to
~12% of the present-day Cenozoic Duero basin drainage area. Such a reduction of the drainage
area could have strong implications on the evolution of the Duero basin, as important lowering
of water and sediment fluxes, and so of incision throughout the basin. To better resolve the
impact of such drainage area reduction on incision capacity, we perform a stream power analysis
of the Duero river. We consider the specific stream power, $\omega$, defined as $\omega = \rho\, g\, Q\, S\, /\, W$, where
$\rho$ is water density, $g$ is gravitational acceleration, $Q$ is discharge, $S$ is local river gradient, and
$W$ is river width (see the Supplement for details of the calculation). We calculate $\omega$ for the
present-day Duero river, and for a restored ancient Duero river that drained this 12% of lost
area. We plot the difference (ancient – present day) between the two curves in Figure 10, with
the Duero river long profile. Calculated difference in specific stream power values are relatively
low ($< 2$ W m$^{-2}$) for the upstream part of the basin, but increase to ~5 W m$^{-2}$ when approaching
the major knickzone at a distance of ~350 km from the river mouth. The knickzone is
characterized by peak values exceeding 10 W m$^{-2}$, which rapidly decrease to ~0 W m$^{-2}$ at the
base of the knickzone (~200 km) and up to the river mouth (Fig. 10). Some alternating peak
and null values are observed in the lower reach of the river and may be related to the occurrence
of numerous dams along the river. Overall, the specific stream power calculated for the ancient
Duero river show higher values than for the present day from the base of the knickzone to the
uppermost reach of the river (Fig. 10). This implies a general decrease of the Duero river's
incision capacity between this ancient state to the present day, magnified on the knickzone.

3.4 $\chi$ map

The comparison of the shape of longitudinal profiles of rivers across divide is a way that has
been proposed recently to infer disequilibrium between rivers and the potential migration of
their divide (Willett et al., 2014). The $\chi$-analysis of river profiles (Perron and Royden, 2012) is
a powerful tool to evidence differences in the equilibrium state of rivers across divide, and then
to infer their migration (Willett et al., 2014). This method is based on a coordinate
transformation allowing linearizing river profiles (Perron and Royden, 2012). Considering
constant uplift rate (U) and erodibility (K) in time and space the χ-transformed profile of a river
is defined by the following equation (Perron and Royden, 2012; Mudd et al., 2014):

$$z(x) = z_b(x_b) + \left(\frac{U}{KA_0^m}\right)^{1/n} \chi$$

with
$$\chi = \int_{x_b}^{x} \left(\frac{A_0}{A(x)}\right)^{\frac{m}{n}} dx$$
where z (x) is the elevation of the channel, x is the longitudinal distance, $z_b$ is the elevation at
the river's base level (distance $x_b$), A is the drainage area, $A_0$ is a reference drainage area, and
exponents m and n are empirical constants.

When using the χ variable instead of the distance for plotting the elevation z along channel, (χ-
plot), the longitudinal profile of a steady-state channel is shown as a straight line (Perron and
Royden, 2012). Any channel pulled away from this line is in disequilibrium and is then expected
to attempt to reach equilibrium. Mapping χ on several watersheds and comparing χ across
drainage divides is then a potential way to high disequilibrium between rivers across divide and
to elucidate divide migration and drainage reorganization through captures (Willett et al., 2014).
We used the χ-analysis Tool developed by Mudd et al. (2014) to select the best m/n ratio by
iteration (Perron and Royden, 2012) and to calculate χ for rivers throughout the divide between
the Ebro and Duero basins from a similar base level at 850 m a.s.l. The best mean m/n ratio for
all our streams is 0.425, which falls in the typical range of values observed for rivers (~0.4 –
0.6: e.g. Kirby and Whipple, 2012). The resulting map (Fig. 11) shows χ values calculated on
different opposite streams in the vicinity of the Ebro/Duero drainage divide. Similar values on
both sides of the divide suggest the two opposite streams are at equilibrium, whereas strong
contrasted χ values imply disequilibrium leading to divide migration, continuously or through
fluvial capture, toward the high χ values (Willett et al., 2014).The map of χ values actually
shows significant contrasting values across the Ebro/Duero divide. We comment here these
contrasts along the divide from the SE to the NW of the area considered (Fig. 11).
There is a strong contrast in χ values between the headwater of the Jalon river (Fig. 11),
characterized by low values (~300 m), and the closest part from the divide of the Bordecorex
river (Fig. 4), a tributary of the Duero river (~500 m). Such a disequilibrium implies divide
migration toward the Duero basin, predicting the capture of the uppermost reach of the

Bordecorex river by the Jalon river. To the north, tributaries of the Jalon river show slightly lower χ values than the tributaries of the Duero river. This suggests a relative stable situation although small captures may occur toward the Duero basin. A higher contrast is observed around the easternmost part of the Duero basin, which is surrounded by the Ebro basin. The Araviana river (tributary of the Duero river) seems to be taken in a bottleneck between the Manubles river to the south and the Queiles river to the north (Fig. 4), which both show lower χ values (Fig. 11). Toward the east, there is a strongest χ values contrast between headwaters of the Araviana river (>700 m) and of the Isuela (Jalon tributary) and Huecha rivers (<100 m). This domain appears clearly in disequilibrium and is expected to be captured by the Ebro drainage network. Such high χ values differences appear also to the northwest (Fig. 11), in the southern part of the Cameros basin where the Duero river and its tributaries' headwaters show χ values >500-700 m, whereas the facing rivers (Alama, Cidacos, Iregua, and Najerilla) are all characterized by low χ values <100 m. This predicts important disequilibrium and divide migration and fluvial captures toward the south. Northwestward, χ values between Duero and Ebro network are more similar indicating that the divide is relatively more stable here, up to the westernmost part of the Ebro basin (Fig. 11). However, there are some slight localized χ value contrasts (~200 / ~450 m) as observed between the Tiron and the Arlanzon rivers, between the Rudron and the Ubierna and Urbel rivers, and between the Ebro and the Pisuerga rivers (Fig. 11). It suggests minor local captures toward the Duero basin.

To sum up, χ values calculated in the vicinity of the drainage divide between the Ebro and Duero river networks show a general disequilibrium (Fig. 11) as the Ebro network is characterized by low χ values (up to ~200-300 m) compared to those for the Duero network (up to ~450-700 m). In complement with all the evidence of divide displacements induced by captures described previously this allows predicting a general divide migration toward the Duero basin through headwater retreat, in favor of the Ebro tributaries, especially around the Almazan subbasin, which is expected to be entirely captured by the Ebro basin.

**4. Discussion**

4.1 Long term trend of divide migration

The oldest capture evidence in our study area corresponds to the incision of the northern part
of the Iberian Range by the Jalon river and by the capture of the Calatayud basin, attributed to
the post-Messinian (Gutiérrez-Santolalla et al. 1996). We propose, based on morphological
evidence (Fig. 4) and in agreement with stratigraphic data (Gutiérrez-Santolalla et al. 1996),
that the Jalon river system captured the Jiloca graben to the east since the Early Pleistocene,
before progressively capturing the Almazan subbasin toward the west in the Holocene
(Gutiérrez-Santolalla et al. 1996). From $\chi$–analysis (Fig. 11), we deduce that the eastern part of
the Duero basin, the Almazan subbasin, is being actively captured by Ebro tributaries that
drained the Iberian Range and the Cameros basin. Despite low contrasts in $\chi$ values, local
captures are also suggested in the vicinity of the Ebro / Duero drainage divide toward the
northwest. Capture is further implied by the occurrence of numerous high elevated (~1000 m)
knickpoints and low-relief surfaces (Figs. 5, 8, 9, 11).
Thus, there is a good correlation between $\chi$ evidence and morphological and stratigraphic data
implying long-lasting captures and divide migration during Pliocene, Pleistocene, and
Holocene times in favor of the Ebro basin.

The pursuit of such a long-term capture trend may be driven by tectonic and/or climatic forcing
(Willett, 1999; Montgomery et al., 2001; Sobel et al., 2003; Sobel and Strecker, 2003; Bonnet,
2009; Whipple, 2009; Castelltort et al., 2012; Kirby and Whipple, 2012; Goren et al., 2015; Van
der Beek et al., 2016). However, such long-term trend in drainage reorganization may also occur
in tectonically quiescent domains, independently of external forcing (Prince et al., 2011). Here,
the Iberian Range and the Cameros basin recorded extension pulses from the Late Miocene to
the Early Pleistocene, responsible for the formation of several grabens as previously described
(Gutiérrez-Santolalla et al., 1996; Capote et al., 2002). Extension events are also recorded
during the Holocene, nevertheless, the youngest erosion surface of Late Pliocene-Early
Pleistocene age observed in our study area shows no tectonic-related deformation and
reworking, suggesting that tectonic activity is reduced here (Gutiérrez-Elorza and Gracia,
1997). This is also consistent with the relative scarcity of seismic activity observed in our study
area, compared, for instance, to the Pyrenees, or to the Betics (Herraiz et al., 2000; Lacan and
Ortuño, 2012). We consequently propose that local tectonic activity is not the main driver of
the capture histories documented here, as most capture events postdate the cessation of tectonic
activity, and occur during periods of quiescence (Gutiérrez-Santolalla et al., 1996).

The Cameros Massif if characterized by relatively high mean annual precipitation up to ~1000
mm/an (Fig. 6) with high elevation (~1400-2200 m) in comparison with the surrounding areas.
This contrasts with the adjacent Ebro and Duero basins where low precipitation rates, of ~400-
500 mm/an (Hijmans et al., 2005), illustrate subarid climate conditions. The Cameros area is
the only place in our study area where a contrast in precipitation pattern (Fig. 6) would
potentially drive a migration of the divide toward the drier, Duero area. Given that the same
pattern is observed everywhere, even where there isn't any precipitation difference, we suggest
that the present day climatic condition is unlikely to control the general pattern of current
drainage reorganization between the Ebro and Duero basins. During the Pliocene and the
Pleistocene, the climatic record in the northern Iberia Peninsula is characterized by alternations
between similar subarid conditions and intense glaciation. Paleoclimate proxies do not allow to
highlight past precipitation differences along the divide that could explain past drainage
reorganization. Moreover, there is no clear evidence of important glacier development and
related erosion in our study area, especially for the Cameros basin and the Iberian Range
(Ortigosa, 1994; García-Ruiz et al., 1998, 2016; Pellicer and Echeverría, 2004). This indicates
that drainage evolution between the Ebro and Duero basins is unlikely to be related to climatic
evolution.
4.2 Excavation of the Ebro basin as the main factor controlling divide migration and limiting
incision of the Duero river
A striking morphological feature for river capture in our study area is the important contrast in
the incision pattern (e.g. Fig. 1B) from one side of the divide to the other. This suggests that the
incision capacity of the river network is the main driver for capture and divide migration. Both
tectonic and climatic forcing does not appear to control drainage reorganization between the
Ebro and Duero basins.
The opening of the Ebro basin toward the Mediterranean Sea during the Late Miocene led to
widespread excavation (Garcia-Castellanos et al., 2003, Garcia-Castellanos and Larrasoaña,
2015), favored by more humid and seasonal climatic conditions (Calvo et al., 1993; Alonso-
Zarza and Calvo, 2000). By contrast, incision related to the opening of the Duero basin toward
the Atlantic Ocean is concentrated to the west in the Iberian Massif, characterized by a
largescale knickzone (150 km long and 500 m high) in the Duero river long profile (Fig. 1B).
This contrasts with the limited eastward propagation of incision in the Cenozoic part of the
basin (Antón et al., 2012, 2014), despite climatic conditions similar to the Ebro basin. An
explanation resides in the fact that the resistant Iberian Massif basement rocks may have
controlled and limited incision and drainage reorganization in the Cenozoic Duero basin (Antón
et al., 2012). The Duero profile upstream of this major knickzone may be considered as a high
elevated local base level for its tributaries there. Difference between the Ebro and Duero base-
levels implies a major contrast in fluvial dynamics. We suggest the systematic and long-term
trend of divide migration toward the Duero basin and fluvial capture in favor of the Ebro basin
is driven by the differential incision behavior, controlled by base-level difference.
Our stream power analysis along the Duero river (Fig. 10) shows that the difference in drainage
area of the Duero inferred from our paleo-divide map (Fig. 9) induces a noticeable decrease of
stream power values of the Duero in the vicinity of the knickzone. This stream power is a
minimum estimate because calculation does not take into account possible captures and divide
migration in other areas along the Duero basin divide, nor the full history of the divide migration
through time and the related ongoing decrease in water discharge as documented in laboratory-
scale landscape experiments (Bonnet, 2009). Some contrasts of incision are also observed in
the Iberian Range along the southern border of the Duero, and in the Cantabrian domain to the
North. Both show more important incision than in the Duero basin, suggesting potential river
captures and divide migration at the expense of the Duero basin, increasing the total of lost
drainage area.  Even if it gives minimal estimate, our stream power analysis suggests that
drainage area reduction may have limited the erosion in the Duero basin. This provides an
explanation for the preservation of the lithologic barrier to the west, along the main knickzone
of the Duero considered as an intermediate, local base level (Antón et al., 2012).We propose
that the reduction of the Duero drainage area caused by captures and incision in the Ebro basin,
is responsible for a significant decrease of the incision capacity in the Duero basin. We infer
that the ongoing drainage network growth in the Ebro basin may be responsible for the current
preservation of large morphological relics of the endorheic stage in the Duero basin.
The opening of the Ebro basin toward the Mediterranean Sea resulted in a drastic base level
drop. This results in the establishment of an upstream-migrating incision wave that propagates
to every tributary of the Ebro network, responsible for knickpoints migration (Schumm et al.,
1987; Whipple and Tucker, 1999; Yanites et al., 2013) and for drainage reorganization and
divide migration. The χ-analysis that we performed along the current Ebro-Duero divide (Fig.
11) highlights areas where geomorphic disequilibrium is still ongoing, which suggests that they
are areas where divide is currently mobile. The modelling study performed by Garcia-
Castellanos and Larrasoaña (2015) suggests that the re-opening of the Ebro basin occurred
between 12.0 and 7.5 Ma. This indicates that the growth of the drainage network of the Ebro

basin and the establishment of new steady-state conditions is a long-lived phenomenon, which is still not achieved today.

**Conclusion**

In this paper we present a morphometric analysis of the landscape along the divide between the Ebro and Duero drainage basins located in the northern part of the Iberian Peninsula. This area shows numerous evidence of river captures by the Ebro drainage network resulting in a long-lasting migration of their divide, toward the Duero basin. Although these two basins record a similar geological history, with a long endorheic stage during Oligocene and Miocene times, they show a very contrasted incision and preservation state of their original endorheic morphology. Since the Late Miocene, the Ebro basin was opened to the Mediterranean Sea and record important erosion. On the opposite, the Duero was opened to the Atlantic Ocean since the Late Miocene – Early Pliocene but its longitudinal profile exhibits a pronounced knickpoint, which delimits an upstream domain of low relief and limited incision, likely representing a relict of its endorheic topography. We propose that this contrast of incision is the main driver of the migration of divide that we document. The morphological analysis of rivers across the divide highlights areas where geomorphic disequilibrium is still ongoing, which suggests that the Ebro-Duero divide is currently mobile. The quantification of the decrease of the drainage area of the Duero based on the reconstruction of a paleo-position of the Ebro-Duero divide shows that the divide migration results in a significant lowering of the stream power of the Duero river, particularly along its knickzone. We suggest that divide migration induces a decrease of the incision capacity of the Duero river, thus favoring the preservation of large relicts of the endorheic morphology in the upstream part of this basin.

Author contributions

AV undertook morphometric modeling and interpretation, and wrote the paper. SB and FM contributed to the interpretation and the writing.

Competing interests.

The authors declare that they have no conflict of interest.

Acknowledgements.
This study was funded by the OROGEN Project, a TOTAL-BRGM-CNRS consortium. We
thank two reviewers and associated Editor Veerle Vanacker for very useful and constructive
comments that greatly helped us to clarify and improve this manuscript.

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

Figure captions:

Figure 1: A) Topographic map of the Duero and Ebro basins and surrounding belts. B) Averaged topographic section throughout the Duero and Ebro basins showing important incision contrast between the two basins. The Duero basin recorded low incision, especially in its upper part, whereas the Ebro basin is highly excavated.

Figure 2: Simplified geological map of the study area.

Figure 3: Topographic map of the study area with all the rivers considered in this study. The red lines represent drainage divides between main hydrographic basins.

Figure 4: Zoom in the geological map of the Iberian Range showing the location of the Jalon river tributaries. The river long profiles of these streams and the location of knickpoints are shown to the left.

Figure 5: A) Zoom in the geological map of the Bureba sector. B) Zoom in the Homino river (Ebro tributary) capturing the upper reach of the Jordan river (Duero tributary). C) Schematic representation of this capture using river long profiles and map orientation, showing the associated knickpoint and wind gap.

Figure 6: Mean annual precipitation map for the study area (data from Hijmans et al., 2005).

Figure 7: A) 3D view of the DEM of the Bureba sector showing important contrast of incision between the Ebro and Duero basins across their divide (red dashed line) and river capture evidence (elbows of capture, knickpoints and wind gaps). B) Google Earth image around the locality of Hontomin where the Homino river is capturing the upper reach of the Jordan river. C) and D) Wind gaps cut into the Bureba anticline (see location on Fig. 7A). Pictures have been taken from the north of this structure toward the south. E) Possible three steps evolution of the southwestward divide retreat through multiple river captures witnessed in the area.

Figure 8: River long profiles for all the streams described in the Bureba area showing evidence of river capture. Colors are given to rivers that are linked in these capture processes.

Figure 9: Topographic map showing the location of all the knickpoints and low relief surfaces that may be associated to river capture. The black dashed line represents a possible paleodrainage divide between the Ebro and Duero basins. The area between this dashed line and the present-day location of the divide in red may have belonged the Duero basin before being captured by the Ebro basin.

Figure 10: Duero river long profile (black line) and difference in the specific stream power of the river (grey) calculated by considering the paleo and present-day position of its divide. Positive values suggest a significant diminution of the incision capacity of the Duero river, particularly along the knickzone of its longitudinal profile. Details on calculation are available in the Supplement (Section S1).

Figure 11: Topographic map with $\chi$ values calculated on different opposite streams in the vicinity of the Ebro/Duero drainage divide. This map shows significant contrasting values between the Ebro and Duero drainage networks.

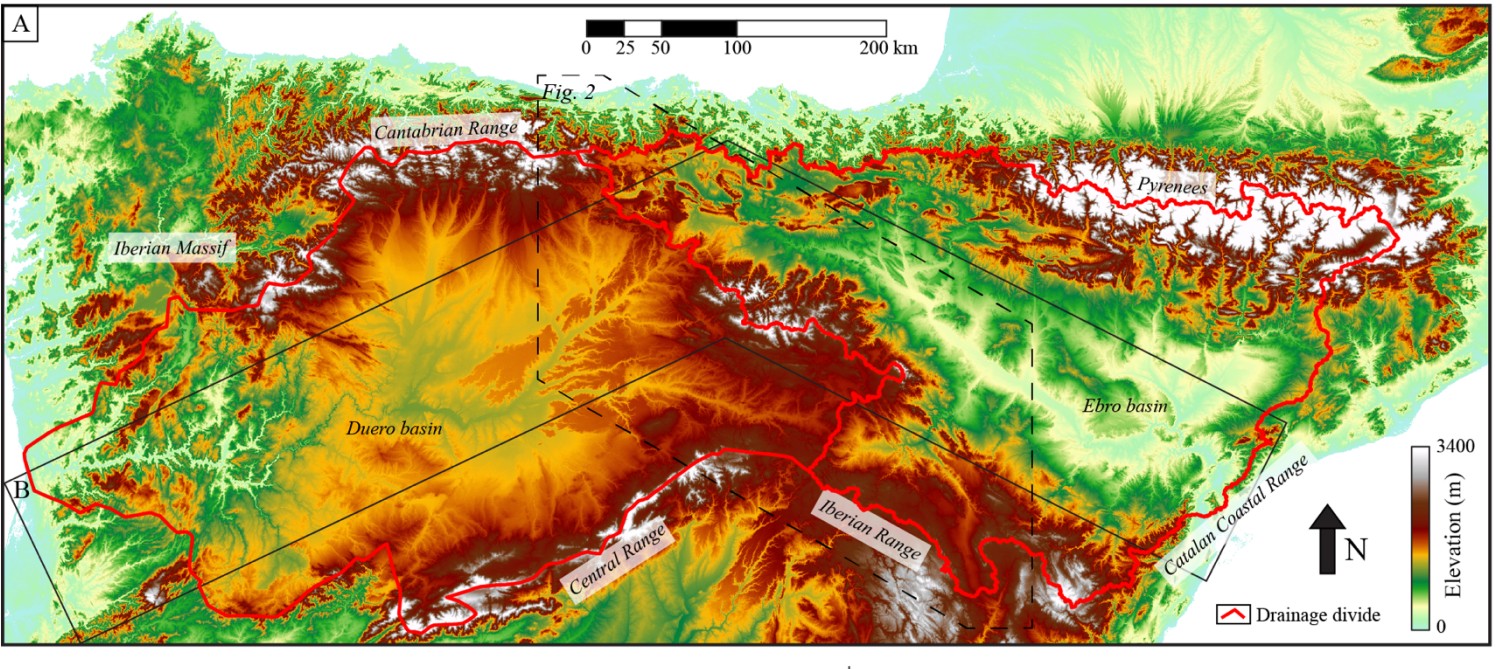

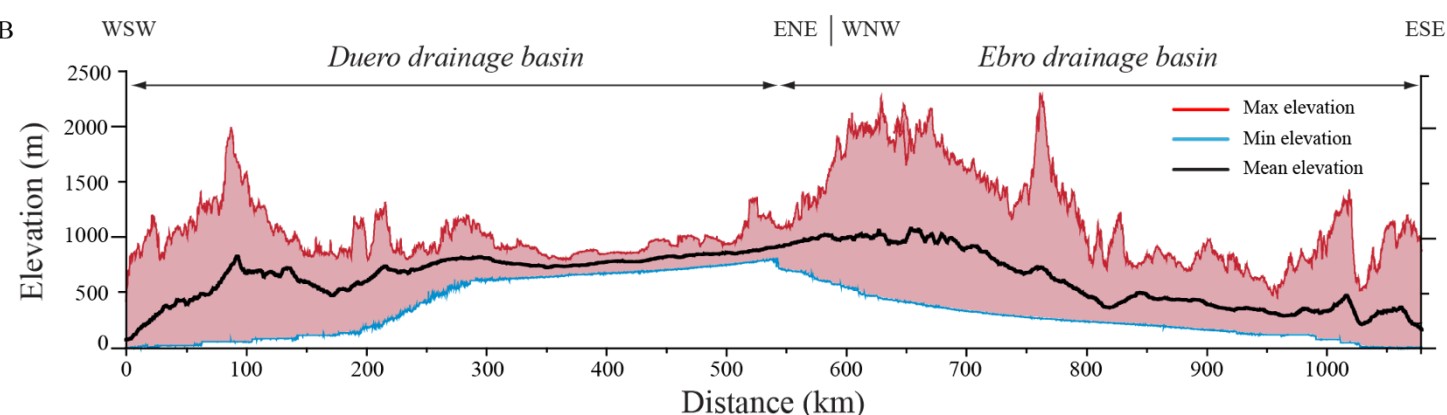

Figure 1

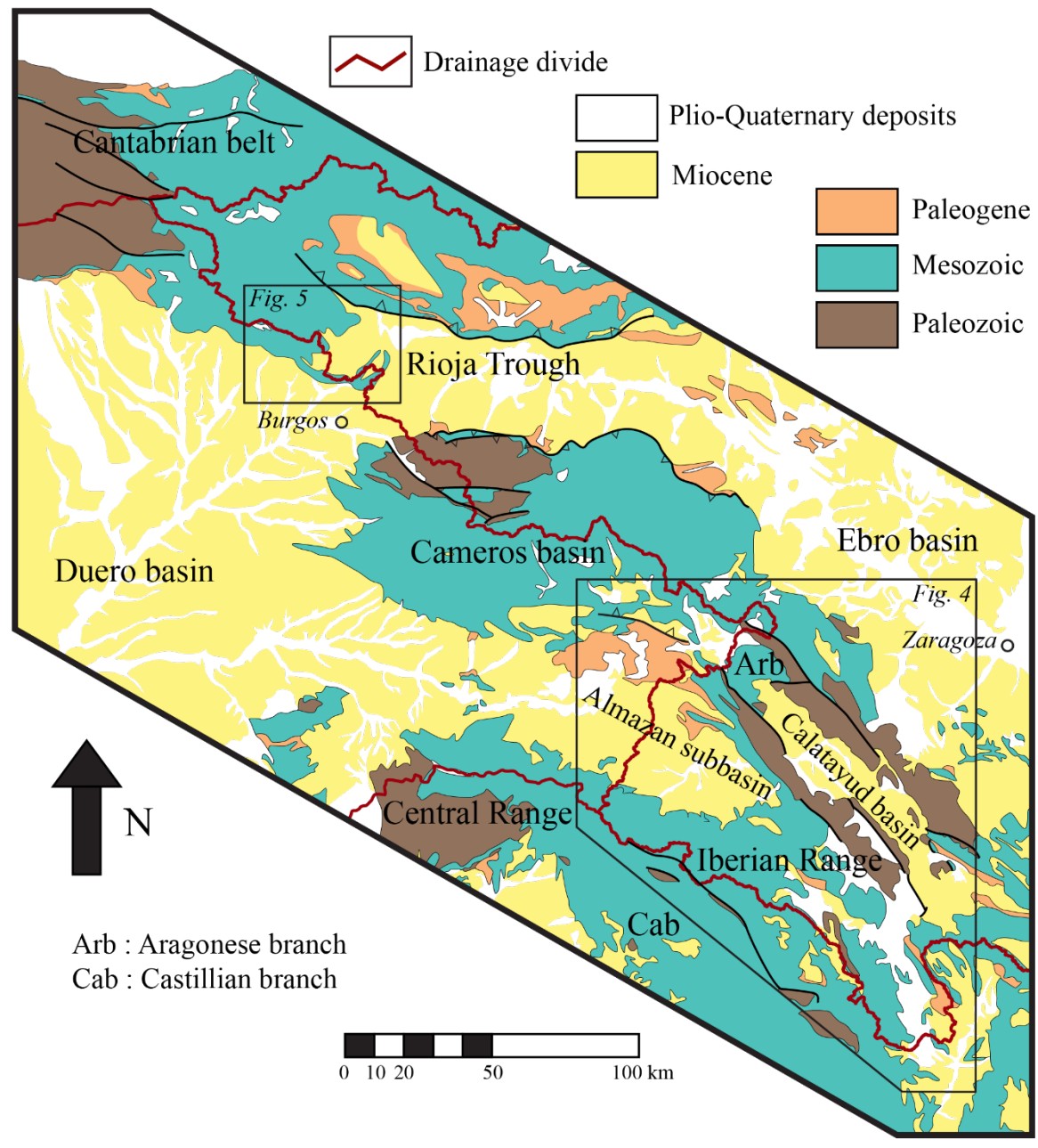

Figure 2

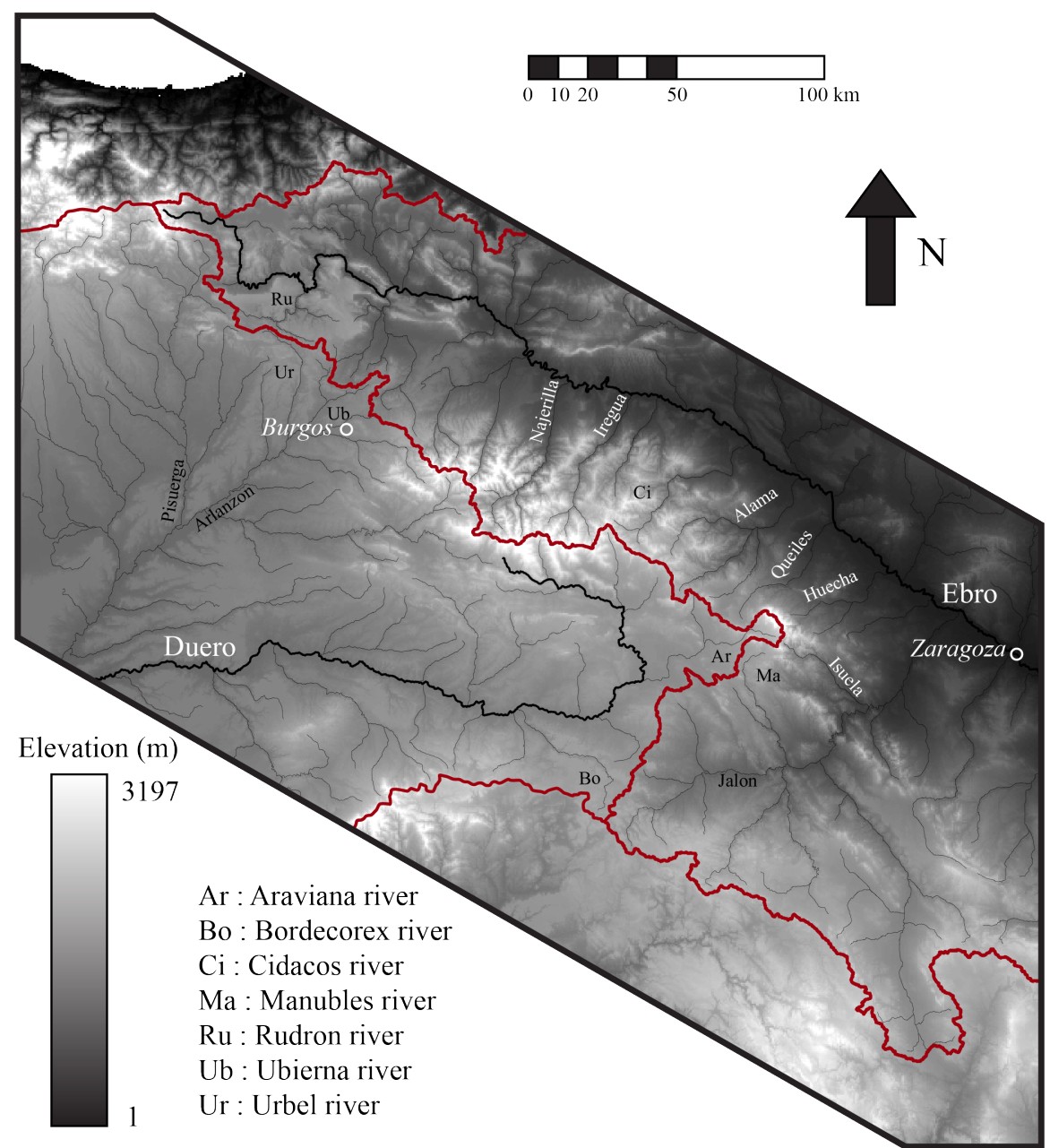

Elevation (m)

3197

Ar : Araviana river
Bo : Bordecorex river
Ci : Cidacos river
Ma : Manubles river
Ru : Rudron river
Ub : Ubierna river
Ur : Urbel river

Figure 3

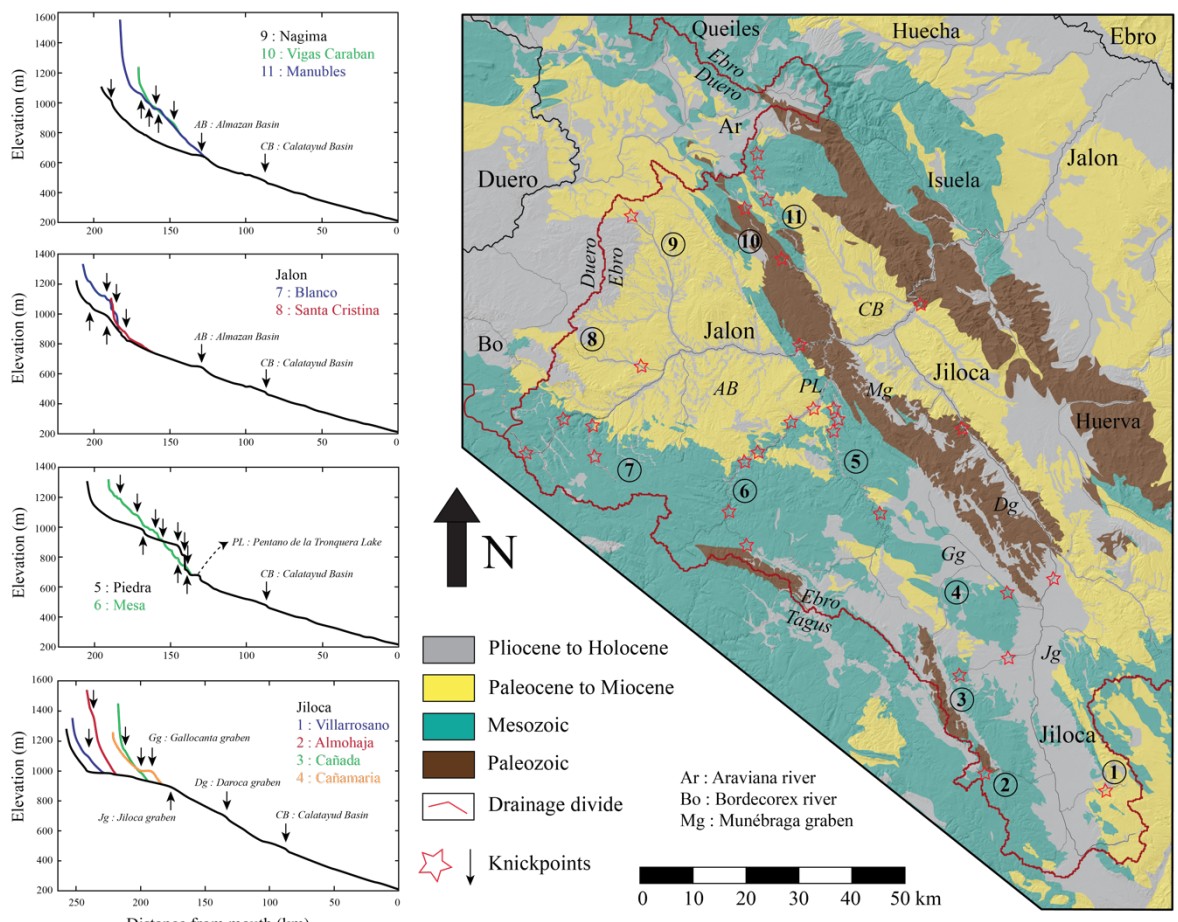

Figure 4

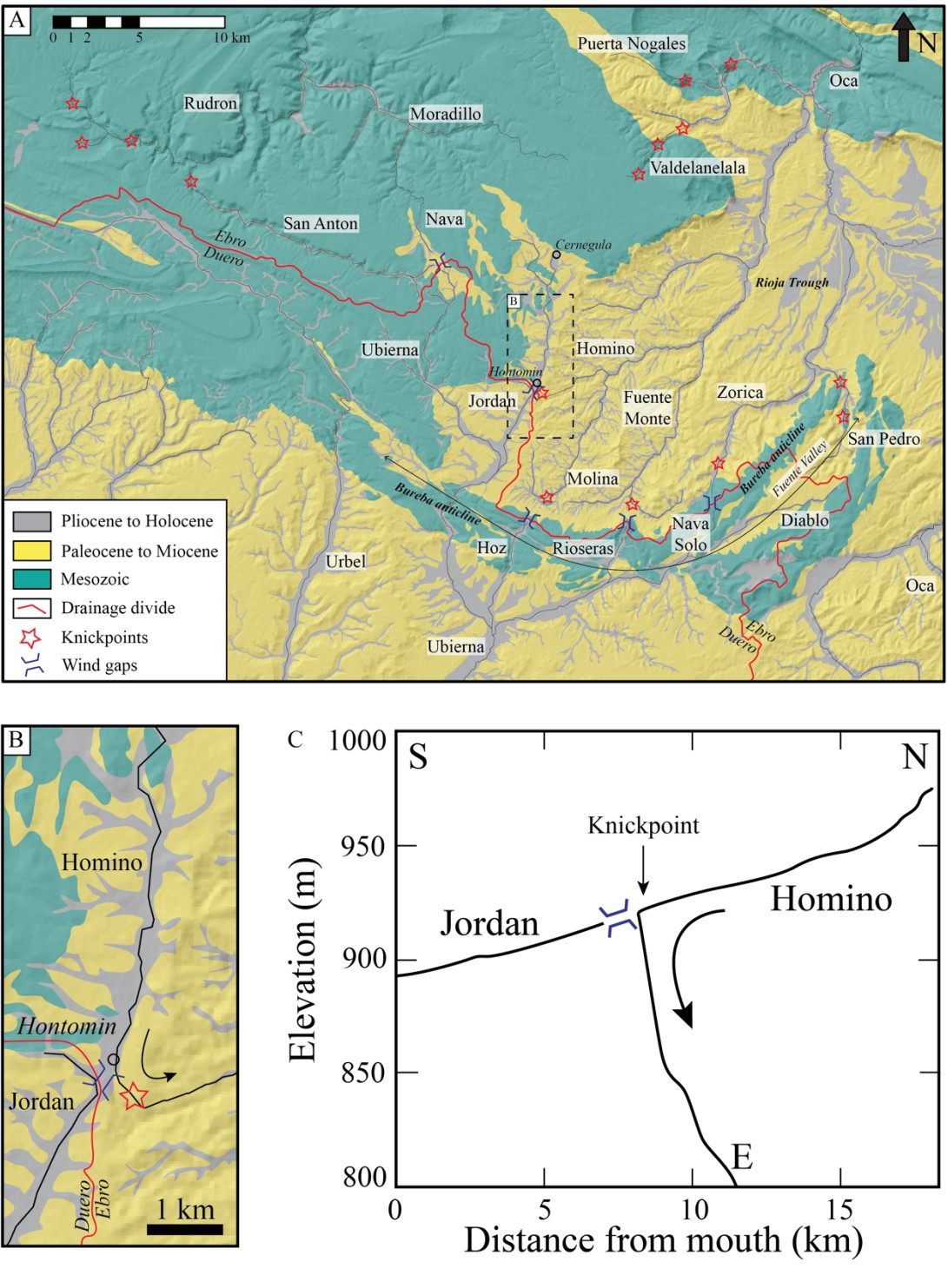

Figure 5

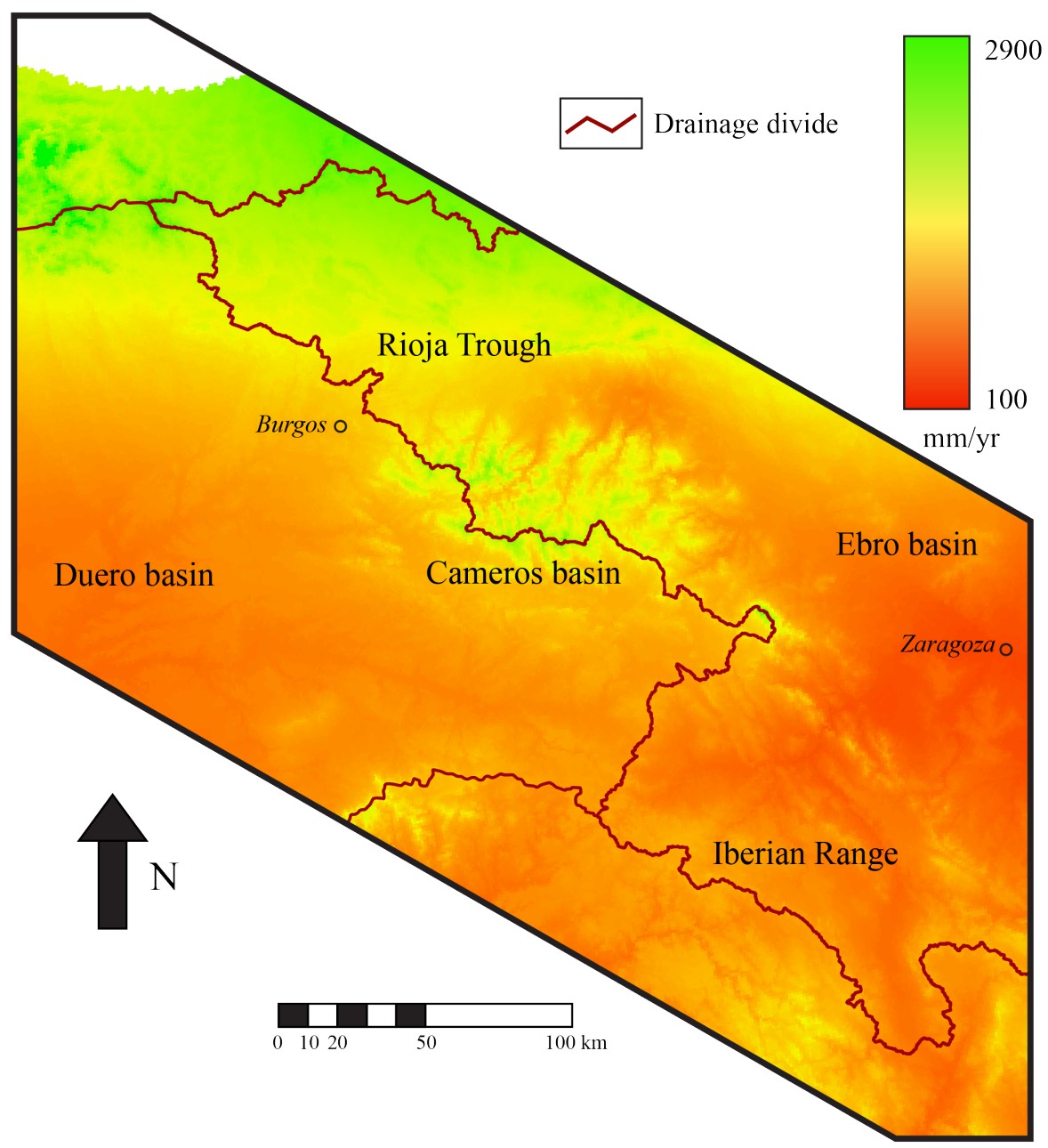

Figure 6

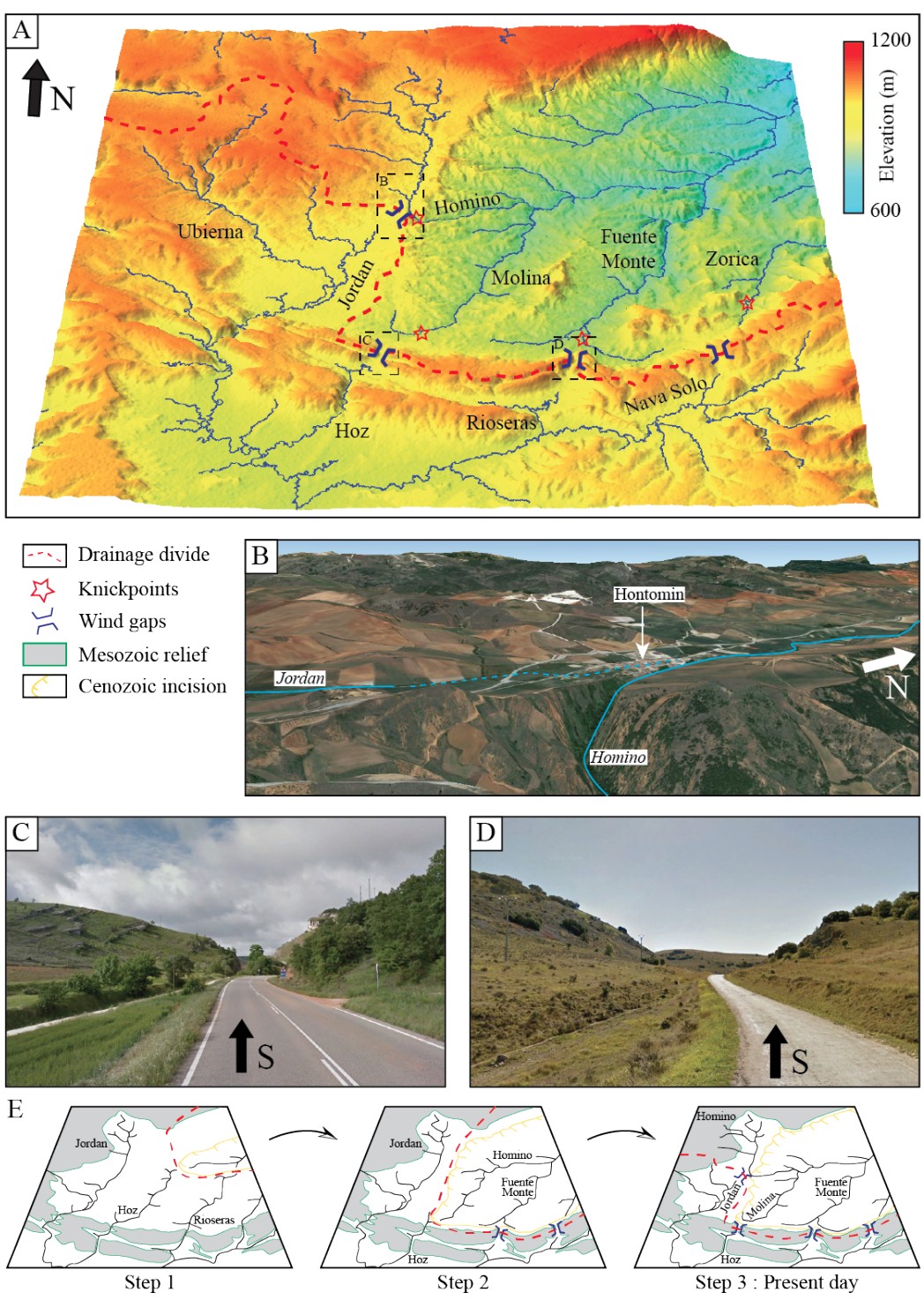

Figure 7

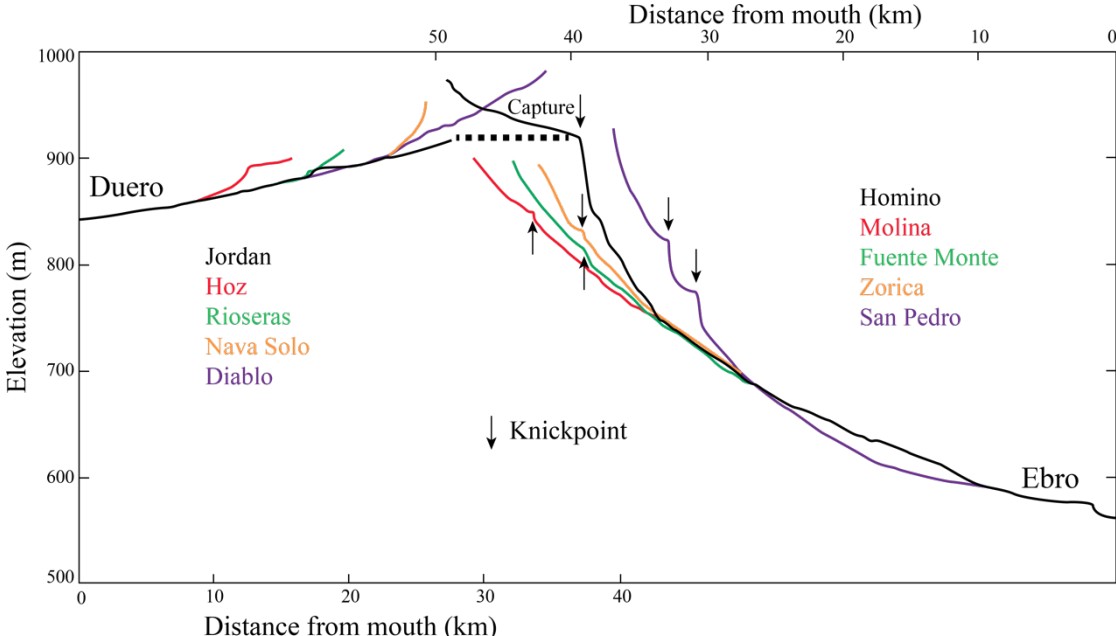

Figure 8

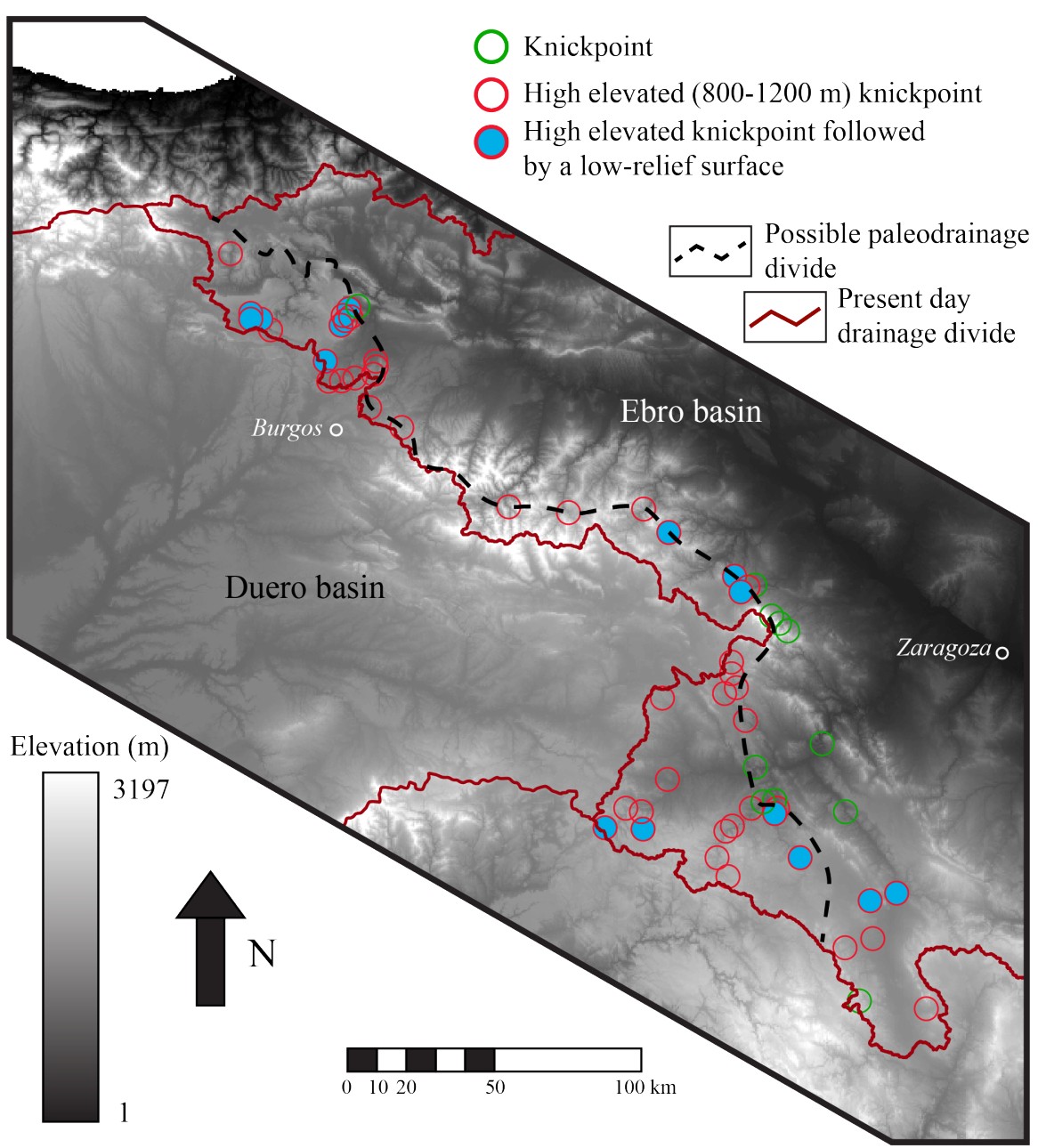

Figure 9

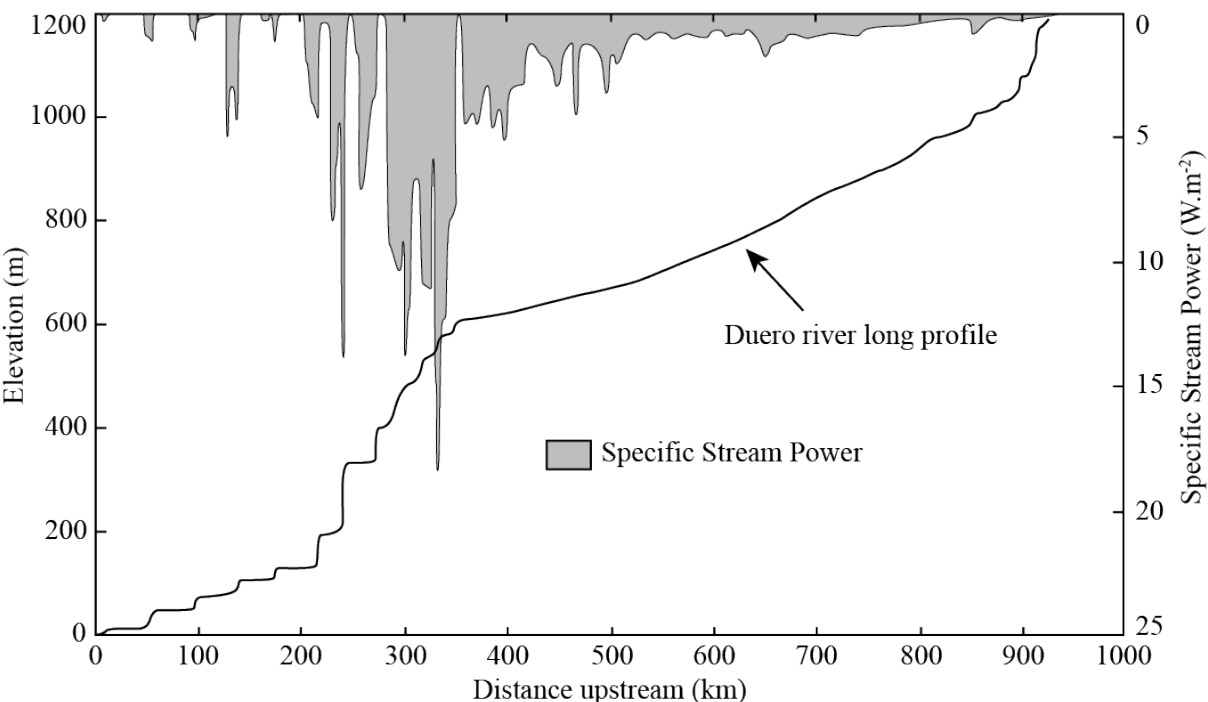

Figure 10

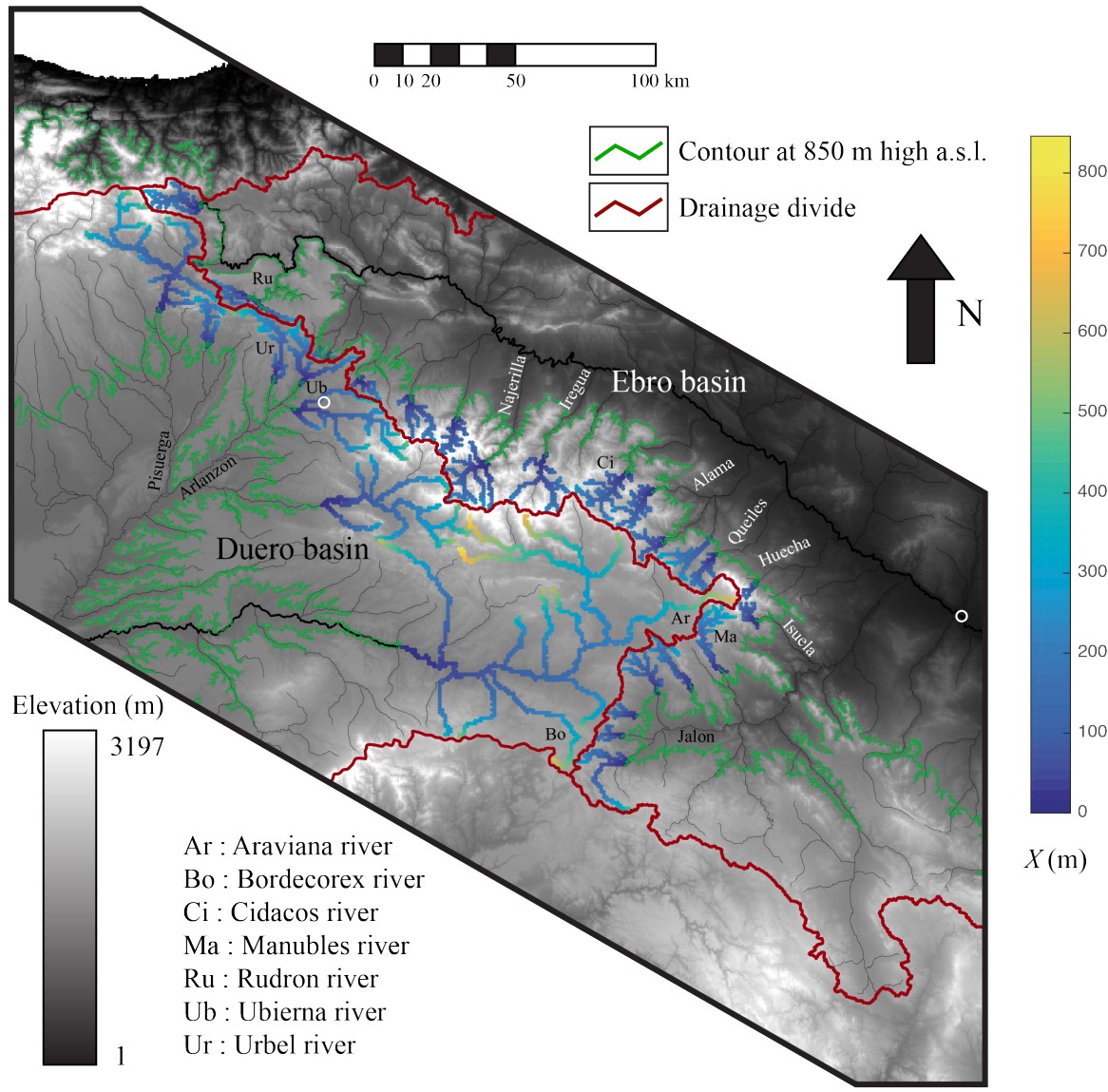

Figure 11