# Peer review of "Arnaud Vacherat1, Stéphane Bonnet1, Frédéric Mouthereau1"

_Earth Surface Dynamics, 2017_

## Referee Comment (RC1) · Anonymous Referee #1 · 20 Oct 2017

The authors make use of a standard topographic DEM to find knickpoints and identify potential drainage divide migrations in the Ebro-Duero catchment boundary. The subject is of interest for the understanding of divide migration and the ms. would spread a geomorphological region that is not so well documented.

The title is appropriate and the abstract responds to the contents, though the abstract could be more direct and comprehensive.

The authors explore a number of captures, most prominently the Homino River one. This one is well known to geomorphologists, and documented not only by Mikes 2010, but also for example in the report accompanying the corresponding chart of the Ge-

ological Map of Spain, published by the IGME. I also recall public panels on display for the random visitor in the Hontomin village, explaining the fluvial captures. I'm sure further bibliographical research will bring even more appropriate references.

I counted 5 mentions of the Duero basins as being "almost still endorheic". The expressions used are inappropriate and misleading. The entire Duero Basin is exorheic today. Perhaps the authors mean that the top of the sedimentary infill is relatively well preserved, and incision is small/recent relative to the Ebro. This needs clarification because it seems to be central point of the article.

647-650: The logics behind this reasoning are obscure. Again, what is a "quasi-endorheic configuration"? And how does the Ebro piracy effect on it? In terms of drainage area, the Duero is still larger than the Ebro basin, so area is not the problem.

Some formal aspects need attention: two entire paragraphs of the abstract are copied as such in the main text, f.e., the last par. in the Introduction. Instead of the latter, I would expect an explanation of what is to come later in the paper, what question is addressed and what strategy they follow. Also the first half of the conclusions are not conclusions. Lines 415-454 are really redundant because that derivation is shown in many earlier papers, and is not relevant to the paper. I would highlight Perron & Royden, ESPL, 2013, as the original authors. Point "3.2.2" should in my view be the "Results", which seem poor relative to the Discussion. Overall, the ms. at its present stage focuses more on interpretation/speculation than on its new objective results. I don't report here any further on formal issues.

Line 46: A sentence as strong as that needs mentioning Gilbert, 1887; Brocklechurst & Whilpple, 2002; Perron & Royden, 2013; for example. Line 120: Specify the ages proposed for the drainage change by the various authors, otherwise the ongoing subject for discussion is not clear. Line 123: River capture AND sediment colmatation of the basin are the two competing processes proposed in the cited references. 132-135: the use of the word 'opposed' here, and the whole sentence, are unclear. Line 222:

the "dynamically stable" sentence is not understandable to me.

---

## Author Comment (AC1) · 20 Nov 2017

We thank the anonymous referee #1 for his useful comments.

In this study, we prominently focus on captures in the vicinity of the Homino River because it is a place where magnificent river captures are well preserved in the landscape. It is consequently a select area for studying drainage reorganization and the induced divide migration, as already also recognized by Mikes (2010). Mikes (2010) already did substantial work in this area and we referred to his publication several times in the submitted ms. We did not know the report accompanying the Geological Map of Montorio, published by the IGME (1997) (now referred to as Pineda (1997) in the

revised ms) and we are pleased to notice that the reconstruction that we proposed (independently) in our Figure 7 (steps 2 and 3) is consistent with what he actually proposed 20 years ago. We added the reference to this work in the revised ms. Note that in the submitted ms, this is not the only study that has mentioned some capture events in our studied area (Gutierrez-Santolalla et al., 1996).

Indeed, the Duero Basin is exorheic as it is opened toward the Atlantic Ocean. However, the upstream part of this hydrographic basin, west of the Iberian Massif is so well preserved, and recorded so low incision, that this domain appears at least to partially preserve the endorheic configuration of the basin. That is why we use the expression: "almost still endorheic". We agree with the reviewer that this expression could be misleading and we use consequently the term "relict of the endorheic stage" in the revised ms. We clarify this in the main text.

L.647-650: The Ebro piracy led to important river capture at the expense of the Duero Basin. The Duero basin then recorded a loss of drainage area through time. This is not a question of absolute size of the Duero drainage basin but a question of decrease of its drainage area. We propose that the decrease of its area, estimated here to be of $\sim$12% on the basis of existing markers, resulted in a decrease of its incision capacity. It is the reason why we consider that the Ebro piracy is responsible for the preservation of large relicts of the endorheic configuration of the Duero Basin. We rephrased the sentences mentioned by the reviewer:

"We then suggest that the decrease of drainage area of the Duero basin that we document here, partly due to divide migration induced by capture and incision in the Ebro basin, is responsible for an important decrease of the incision capacity in the Duero basin. Then, active exorheism in the Ebro basin is likely responsible for the current preservation of large relicts of the endorheic configuration of the Duero basin."

We modified our Introduction to avoid redundancy with the Abstract. We also reorganized it to better highlight the objectives and the tools used in this study.
We also modified the conclusions in order to be more relevant.

L.415-454: even if "redundant" with original derivations of the Stream Power Law, we think necessary to recall this theoretical background in the text to help readers that are not familiar with these methods. This is a very common practice in scientific papers to give some background, even if the authors are not those who developed it initially. We agree that the "Results" section is short compared to the Discussion. However, we consider here our objective results are not limited to the application of the Chi in our study area but also to its combination with geological and geomorphological observations, to finally highlight an original relation between the Ebro excavation and the Duero preservation, and this needed to be discussed.

L.46: We disagree. The reference to Bonnet (2009) is appropriate as this paper exactly deals with what is mentioned in this sentence. The suggested reference to Brocklehurst and Whipple (glacial erosion and relief production) or Perron and Royden (2013) (development of the integral approach (Khi plots) for the study of longitudinal profiles) are clearly off-topic. We also refer to Willet et al. (2014).

L.120: We specify the ages proposed for the Ebro opening for each reference.

L.123: Amended.

L.132-135: We replaced the term "opposed" by "facing". Rivers are facing each other from one side of the divide to the other.

L.222: We deleted the term "dynamically". We consider this area as stable in term of river mobility.
* * *

---

## Referee Comment (RC2) · Anonymous Referee #2 · 12 Jan 2018

GENERAL COMMENTS This study examines watershed migration as a result of capture between 2 of the largest Iberian drainages – the Ebro and the Duero. I think that the paper adds valuable insights into drivers of migration divide, but that these are currently somewhat lost in the manuscript. I suggest that the paper could be fronted and ended with a stronger abstract, introduction and discussion/conclusions that frame the wider implications of the study. This would include emphasizing why this study is important in terms of wider implications (eg type of capture Bishop 1995; implications for landscape Mather et al 2002,), ? I would argue that the key highlights need to be drawn out more clearly e.g. demonstrating the role of lithology as a limiter to incision (seen on smaller scale in other captures eg Shepherd 1982, Mather 2000). Where this

occurs within the basin will impact on how far any sea-level generated wave of incision can propagate up a basin and thus has wide ranging significance to other studies. This occurs in both the mid Duero and lower Ebro in the coastal ranges. I am not suggesting a major re-write here – more a subtle re-wording of the text to incorporate such points and widen the impact of the study.

SPECIFIC COMMENTS / Line 40, 76, 659 – what does 'almost still endoheric" mean? – you mean it retains the landscape signature? clarify / Line 41 – how does the Ebro enable enhoherism of the Duero? I don't understand / Line 74, 578 – what is 'unfilling'? This is not an established term - do you mean erosion? Incision? / Lines 87-88 is one reason why it is important to know this , but it is sandwiched admidst other information – a more explicit 'why this is important' for the study should be provided from the start (currently this is hidden in the text) / Lines 320-1 Contrasts in sharpness could also be attributed to lithological differences - expand. / Line 597 – you mean the headwaters of the modern Ebro rather than the Duero (as this reads)?  – confusing / Lines 613-618 – so how does this compare to other captures – as these are headward these are incremental – overall drainage has time to adjust rather than mid-basin captures which tend to be more of a sudden impact eg smaller , well documented captures such as Sorbas Basin. Stokes et al 2002 / Some areas could be more succinct (do we really need to know about the Late Cretaceous climate in section 2.4?)

TECHNICAL COMMENTS / Line 71, 129, 189, 282, 284, 293, 520 onwards – use of 'evidences' should be evidence (it can only be singular!)  / Line 72 since should be from / Line 99 on should be of / Line 107 'Since early stage collision, (not of) / Line 108 - ?carbonated alluvial sediments? – I do not understand / Line 119 – as well as what process underlies. . .. / Line 122 – you mean upper Duero – there is much incision below this / Line 137 'in' should be 'of' / Line 156 should be 'accommodated' and Hercynian / Line 158 – this event – what event? Clarify / Line 161, 209 onwards 'deformations' should be deformation / Line 162 –' such as the. . ...' / Line 170 – . ...periods of quiescence. . .. / Line 173 such as fluvial. . ... / Line 203 – took place should be 'existed'

/ Line 207 – 'detritic' not needed / Line 228 – network should be networks / Line 230 associated with (not to) / Line 233 precipitation not precipitations / Line 260 tongues are. . . . . . / Line 300 - knickpoints do not 'witness' capture –they may provide evidence of migrating base-levels which may be associated with capture / Lines 304/5 require references to support these ages / Line 306 – 'never been drained before' – I think you mean never externally drained (it would have been drained ie had drainage)? / Line 322 – why 'for instance' ? / Line 331 – of not in / Line 358, 371, 635 – why 'remarkable' do you mean marked? / Line 389 – witnessed? Do you mean suggested by. . .. / Line 391 –recorded = records / Line 398 with not to / Line 408 a lot = many / Line 487, little = small / Line 500, X value contrasts / Line 523 deduce not deduced / Line 525suggested rather than evidence / Line 526-7 implied by rather than well witnessed by / Line 533 – define long (temporal? Spatial?) / Line 536 – trends in not trend of / Line 550 precipitation not precipitations / Line 559 – alternations between (rather than alternance) / Line 560 glacier not glaciers / Line 574'the to first order' – does not make sense – reword / Line 584 you mean 'climatic conditions similar to the. . ..'? / Line 608 'The present drainage of the . . ..' / Line 637 helps with (not for) / Line 647 Then ? - not needed, remove / Line 655 – record (not recorded) / Line 658 'was open to the Atlantic Ocean from the. . .." / Line 659 records not record

REFERENCES REFERRED TO ABOVE / Bishop, E 1995. Drainage rearrangement by river capture, beheading and diversion. Progress in Physical Geography, 19, 449-473. / Mather, A.E. 2000. Adjustment of a drainage network to capture induced base-level change. Geomorphology, 34, 271-289 / Mather. A.E., Stokes, M. & Griffiths, J.S. 2002. Quaternary landscape evolution: a framework for understanding contemporary erosion, SE Spain. Land Degradation & Management, 13, 1-21 / Shepherd, R. G. 1982. River channel and sediment responses to bedrock lithology and stream capture, Sandy Creek drainage, Central Texas. In: Rhodes, D. D.; Williams, G. P. (eds) Adjustment of the Fluvial System. Allen & Unwin, London, 255-275. / Stokes, M., Mather, A.E. & Harvey, A.M. 2002. Quantification of river capture induced base-level changes and landscape development, Sorbas Basin, SE Spain. In: Jones, S.J. & Frostick, L.E. (eds)

Sediment Flux to Basins: Causes, Controls and Consequences. Geological Society, London Special Publication, 191, 23-35.

---

## Editor Comment (EC1) · V. Vanacker (Editor) · 24 Jan 2018

Dear authors,

We have now received two reviews on your manuscript. They provide useful suggestions for the improvement of your final document.

The most important points are :

- Reviewer#1 and #2 asks for a careful revision of the terminology used in the paper, and more specifically to the use of the terms "endorheic", "unfilling", etc.

- Reviewer#1 is concerned about repeated text. The latter concerns – for example -

the sentences on lines 36-42 and 85-91 that are repeated.

- The organization of the paper needs further attention, and the revised paper needs better separation of methods, results and discussion. This is particularly so for section 3 on "morphometric analyses". It is not very clear if section 3.1 is based on own research, a literature review, or a combination of both. As suggested by Reviewer#1, try to highlight your own research findings, and discuss them in the context of the literature and the theory (in section 4). Also, section 4 (discussion) contains new analyses on the stream power, that would rather belong in section 3.

- The theoretical background of the chi analyses is based on Perron and Royden (2013), and Mudd et al. (2014). You can refer to the literature to avoid repetition, and resume section 3.2.1.

- Reviewer#2 suggests rewording parts of the introduction, discussion/conclusion to highlight the importance of your study, and its wider implications.

The two reviews give more detailed comments that need to be addressed in your rebuttal.

With kind regards, and good luck with the revisions, Veerle Vanacker

---

## Author Response (AR1)

To the Associated Editor of ESurf

Dear V. Vanacker

We are pleased to submit the revised version of our manuscript "Drainage reorganization and divide migration induced by the excavation of the Ebro basin (NE Spain)".

We want to thank you and the two reviewers for very useful and constructive comments that greatly helped us to clarify and improve our manuscript. We have significantly reorganized and shortened the manuscript (594 lines against 663), without changing the scientific results and the conclusion.

In the following we respond to the comments point-by-point, beginning by your comments.

Sincerely yours,

Stéphane Bonnet on behalf of the co-authors

**Associated Editor**

We have now received two reviews on your manuscript. They provide useful suggestions for the improvement of your final document.
The most important points are :
- Reviewer#1 and #2 asks for a careful revision of the terminology used in the paper, and more specifically to the use of the terms "endorheic", "unfilling", etc.

We modified the ms accordingly (see details in the response to reviews). Both reviewers asks for clarification of the use of expressions such as "almost still endorheic" or "quasi-endorheic configuration". They are right, we agree that these expressions were misleading. We used these expressions because as already pointed out by Anton et al. for example (ref in the ms), the upstream part of the Duero basin, upstream its main knickzone, is an area where there is a limited incision. This is an area where the endorheic topography is partially preserved. That is why we use the expression: "almost still endorheic". We agree with the reviewer that this expression could be misleading and we use consequently the term "relict of the endorheic stage" in the revised ms. All the occurrences of these misleading expressions have been changed in the ms.

"unfilling" has been replaced by erosion.

- Reviewer#1 is concerned about repeated text. The latter concerns – for example -the sentences on lines 36-42 and 85-91 that are repeated.

We've worked on the ms to avoid repeated text at the end of the abstract (lines 36-42) and introduction (lines 85-91). These sentences were not needed at the end of the introduction and have consequently been deleted in the revised ms.

- The organization of the paper needs further attention, and the revised paper needs better separation of methods, results and discussion. This is particularly so for section 3 on "morphometric analyses". It is not very clear if section 3.1 is based on own research, a literature review, or a combination of both. As suggested by Reviewer#1, try to highlight your own research findings, and discuss them in the context of the literature and the theory (in section 4).

We agree and we reorganized the ms in order to better distinguish between results and interpretations.

We substantially reorganized the section 3 and sub-sections. We have simplified the section 3 (dedicated to results) by creating two new sub-sections, 3.1 and 3.2 where we now present all the evidence for captures in the two studied area, the Iberian Range (3.1) and the Rioja trough (3.2). In the previous ms, some evidence of captures published in the literature were presented first in the section 2 (dedicated to the setting) and then again in the former section 3.1. These sentences (lines 186-195 and 213-223 in the former ms) have been moved and merged with the former section 3.1, in the new sections 3.1 and 3.2 (these changes are highlighted by colors in the revised ms with track changes). Now, section 2 focuses only on the geological setting.

We rewrote the section dedicated to captures (new sections 3.1 and 3.2) by introducing first the evidence of captures from the literature and then by presenting new evidence based on our analysis. We made the choice to present literature and our results in the same sub-sections (instead of presenting all the literature in a sub-section "geomorphological setting" in the setting in section 2 for example) because when we present our results we frequently compared them with the previous work. So we think better, because easier to understand, to have the literature and then our results in the same sub-section.

Finally, we have also added a new sub-section 3.3 where we present the map that shows the paleo-position of the Ebro-Duero divide deduced from all the evidence of captures.

Also, section 4 (discussion) contains new analyses on the stream power, that would rather belong in section 3

We agree. We have moved the section on the stream power analysis to the result section (in the sub-section 3.3) because the calculation is based on the map presented in the result section. Now there isn't any new analyses and results in the section 4 (discussion)

- The theoretical background of the chi analyses is based on Perron and Royden (2013), and Mudd et al. (2014). You can refer to the literature to avoid repetition, and resume section 3.2.1.

Done. The Chi-analysis is now in a new dedicated sub-section 3.4. As suggested by reviewers we have significantly shorten the presentation of the theoretical background and kept only necessary information.

- Reviewer#2 suggests rewording parts of the introduction, discussion/conclusion to highlight the importance of your study, and its wider implications.

Done. We have modified the introduction, discussion and conclusion in that way. We particularly rephrased the conclusion, to better highlight our main findings (see also response to review #1).

The two reviews give more detailed comments that need to be addressed in your rebuttal.

**Review#1**

The authors explore a number of captures, most prominently the Homino River one. This one is well known to geomorphologists, and documented not only by Mikes 2010, but also for example in the report accompanying the corresponding chart of the Geological Map of Spain, published by the IGME. I also recall public panels on display for the random visitor in the Hontomin village, explaining the fluvial captures. I'm sure further bibliographical research will bring even more appropriate references.

The capture in the vicinity of the Homino River is indeed an outstanding example of the many captures that occurred near the Duero-Ebro divide. We decided to show this nice and clear example on figure 7, with some others that exist in the same area. Mikes (2010) already did substantial work in this area and we referred to his publication several times in the submitted ms. Despite our in-depth literature research, we have not found so many publications describing the fluvial captures in our study area (our reference list can attest that we have not only looked at "main-stream" journals). In the first, we were not aware of the report accompanying the Geological Map of Montorio, published by the IGME (1997). In the revised version we have added the reference to this work (Pineda, 1997).

I counted 5 mentions of the Duero basins as being "almost still endorheic". The expressions used are inappropriate and misleading. The entire Duero Basin is exorheic today. Perhaps the authors mean that the top of the sedimentary infill is relatively well preserved, and incision is small/recent relative to the Ebro. This needs clarification because it seems to be central point of the article.

We actually used this expression because of the good preservation of the sedimentary infill formed during the endorheism upstream the major knickzone on the Duero profile, because this area have not been incised. The endorheic topography is relatively well-preserved here because the base-level in this area is the top of the knickzone as already recognized by Anton et al. (ref in the ms). We agree that the expression that we used was possibly misleading so we changed it by "relict of the endorheic stage" in the revised ms. We clarifed this in the main text (cf also our response to associate editor's comments). For example the following sentence (lines 25-26 in the submitted ms):

*"The Ebro basin is highly excavated, whereas the Duero basin is well preserved and may be considered as almost still endorheic."*

has been rephrased (lines 23-24 in the revised ms):

*"The Ebro basin is highly excavated, whereas relicts of the endorheic stage are very well preserved in the Duero basin"*

647-650: The logics behind this reasoning are obscure. Again, what is a "quasiendorheic configuration"? And how does the Ebro piracy effect on it? In terms of drainage area, the Duero is still larger than the Ebro basin, so area is not the problem.
The Ebro piracy led to important river captures at the expense of the Duero Basin. The Duero basin then recorded a loss of drainage area through time. This is not a question of absolute size of the Duero drainage basin but a question of decrease of its drainage area. We propose that the decrease of its area, estimated here to be of ~12% on the basis of existing markers, resulted in a decrease of its incision capacity. It is the reason why we consider that the Ebro piracy is responsible for the preservation of large relicts of the endorheic configuration in the Duero Basin.

We rephrased the sentences (now lines 557-561):

*"We propose that the reduction of the Duero drainage area caused by captures and incision in the Ebro basin, is responsible for a significant decrease of the incision capacity in the Duero basin. We infer that the ongoing drainage network growth in the Ebro basin may be responsible for the current preservation of large morphological relics of the-endorheic stage in the Duero basin."*

Some formal aspects need attention: two entire paragraphs of the abstract are copied as such in the main text, f.e., the last par. in the Introduction.

Done. These sentences have been deleted from the introduction (see letter to Editor)

Instead of the latter, I would expect an explanation of what is to come later in the paper, what question is addressed and what strategy they follow.

Done. We rephrased the last part of the introduction (lines 73-79):

*"Following a presentation of the geological context, we first compile evidence of fluvial captures along the Ebro-Duero divide, based on previous studies and our own investigations, and we map the location of knickpoints and relict portions of the drainage network. We use all these observations to reconstruct a paleo-divide position and to estimate the impact of divide migration in terms of drainage area and stream power. We complement this dataset by providing a map of χ across divide (Willett et al., 2014) to highlight potential disequilibrium state between rivers of the Ebro and Duero catchments. "*

Also the first half of the conclusions are not conclusions.

We agree. The conclusion has been rephrased:

Submitted ms (lines 655-659):
*"The Ebro and Duero basins both recorded a long endorheic stage during Oligocene and Miocene times. Since the Late Miocene, the Ebro basin is opened to the Mediterranean Sea and record important unfilling. This results in important incision driven by a very active drainage network. By contrast, the Duero basin is opened to the Atlantic Ocean since the Late Miocene – Early Pliocene and record only limited incision."*

First part of the conclusion in the revised ms (lines 576-579):
*"In this paper we present a morphometric analysis of the landscape along the divide between the Ebro and Duero drainage basins located in the northern part of the Iberian Peninsula. This area show numerous evidence of river captures by the Ebro drainage network resulting in a long-lasting migration of their divide, toward the Duero basin..."*

Lines 415-454 are really redundant because that derivation is shown in many earlier papers, and is not relevant to the paper.

Done, we significantly reduced the theoretical background (see tracks in the revised ms)

I would highlight Perron & Royden, ESPL, 2013, as the original authors.

We agree. We already referred to this paper in the submitted ms

Point "3.2.2" should in my view be the "Results", which seem poor relative to the Discussion.

We merged sub-sections 3.2.1 (Chi-methodology, significantly reduced) and 3.2.2 (results from Chi analysis) of the submitted ms in a new single sub-section dedicated to the Chi analysis (now 3.4). We also presented some results in the section 3.1 of the former ms (regional map of knickpoints and captures, reconstruction of the paleo-divide position) but actually these findings were not enough separated from literature (see comments in review#2). This has been improved in the revised ms (new sub-sections 3.1, 3.2 and 3.3).

Overall, the ms. at its present stage focuses more on interpretation/speculation than on its new objective results. I don't report here any further on formal issues.

**Review#2**

GENERAL COMMENTS This study examines watershed migration as a result of capture between 2 of the largest Iberian drainages – the Ebro and the Duero. I think that the paper adds valuable insights into drivers of migration divide, but that these are currently somewhat lost in the manuscript. I suggest that the paper could be fronted and ended with a stronger abstract, introduction and discussion/conclusions that frame the wider implications of the study.
We reorganized the ms accordingly to better highlight our main findings.

This would include emphasizing why this study is important in terms of wider implications (eg type of capture Bishop 1995; implications for landscape Mather et al 2002,), ? I would argue that the key highlights need to be drawn out more clearly e.g. demonstrating the role of lithology as a limiter to incision (seen on smaller scale in other captures eg Shepherd 1982, Mather 2000). Where this occurs within the basin will impact on how far any sea-level generated wave of incision can propagate up a basin and thus has wide ranging significance to other studies. This occurs in both the mid Duero and lower Ebro in the coastal ranges. I am not suggesting a major re-write here – more a subtle re-wording of the text to incorporate such points and widen the impact of the study.

We agree that providing evidence of lithology as a limiter to incision would be an interesting topic however we do not think that we provide enough evidence of this mechanism to highlight it in this paper. On the opposite, the role of divide migration in reducing the incision capacity of a river is a phenomenon that, to our knowledge, has never been documented before and we consequently rather preferred to focus our paper on this new topic.

SPECIFIC COMMENTS /
Line 40, 76, 659 – what does 'almost still endoheric" mean? – you mean it retains the landscape signature? clarify
Corrected (see responses to associate editor an reviewer#1)

Line 41 – how does the Ebro enable enhoherism of the Duero? I don't understand
Rephrased (lines 31-35): "*Fluvial captures have strong impact on drainage areas, fluxes, and so on their respective incision capacity. We conclude that drainage reorganization driven by*

*the capture of the Duero rivers by the Ebro drainage system explains the first-order preservation of endorheic stage remnants in the Duero basin, due to drainage area loss, independently from tectonics and climate.*"

Line 74, 578 – what is 'unfilling'?
This is not an established term - do you mean erosion? Incision?
Corrected (erosion)

Lines 87-88 is one reason why it is important to know this, but it is sandwiched admidst other information – a more explicit 'why this is important' for the study should be provided from the start (currently this is hidden in the text)
These lines are about one of the main conclusion of our study and have been removed from the end of the introduction.

Lines 320-1 Contrasts in sharpness could also be attributed to lithological differences - expand.
We add the following sentence (lines 275-276): "*However despite a similar bedrock we cannot ruled out some local influence of the lithology on the shape of these knickpoints*".
"
Line 597 – you mean the headwaters of the modern Ebro rather than the Duero (as this reads)? – confusing
Yes, it was a mistake, Ebro instead of Duero. Corrected

Lines 613-618 – so how does this compare to other captures – as these are headward these are incremental – overall drainage has time to adjust rather than mid-basin captures which tend to be more of a sudden impact eg smaller , well documented captures such as Sorbas Basin. Stokes et al 2002
This is an interesting question that we cannot really address; it would deserve to be investigated for example through a modelling approach
Some areas could be more succinct (do we really need to know about the Late Cretaceous climate in section 2.4?)
We think interesting to present the long-term climatic background because it allows us to propose that the difference in the landscape evolution of the Duero and Ebro is not related to climate

TECHNICAL COMMENTS

Line 71, 129, 189, 282, 284, 293, 520 onwards – use of 'evidences' should be evidence (it can only be singular!)
OK corrected
Line 72 since should be from
OK corrected
Line 99 on should be of
OK corrected
 Line 107 'Since early stage collision, (not of)
OK corrected
Line 108 - ?carbonated alluvial sediments? – I do not understand
Changed: "clastic deposits"
Line 119 – as well as what process underlies. . ..
OK corrected

Line 122 – you mean upper Duero – there is much incision below this
Yes, corrected
Line 137 'in' should be 'of'
OK corrected
Line 156 should be 'accommodated' and Hercynian
OK corrected
Line 158 – this event – what event? Clarify
OK corrected
Line 161, 209 onwards 'deformations' should be deformation
OK corrected
Line 162 –' such as the. . ...'
OK corrected
Line 170 – . . ..periods of quiescence. . ..
OK corrected
Line 173 such as fluvial. . ...
OK corrected
Line 203 – took place should be 'existed'
OK corrected
/ Line 207 – 'detritic' not needed
OK corrected
Line 228 – network should be networks
OK corrected
Line 230 associated with (not to)
OK corrected
Line 233 precipitation not precipitations
OK corrected
Line 260 tongues are. . .. . .
OK corrected
Line 300 - knickpoints do not 'witness' capture –they may provide evidence of migrating base-levels which may be associated with capture
OK corrected
Lines 304/5 require references to support these ages
We add reference to Gutierrez-Santolalla et al., 1996
Line 306 – 'never been drained before' – I think you mean never externally drained (it would have been drained ie had drainage)?
Yes corrected by adding "externally"
Line 322 – why 'for instance' ?
Mistake, changed ("Finally" instead of "for instance")
Line 331 – of not in
OK corrected
Line 358, 371, 635 – why 'remarkable' do you mean marked?
OK corrected (deleted)
Line 389 – witnessed? Do you mean suggested by. . ..
OK corrected (suggested)
Line 391 –recorded = records
OK corrected
Line 398 with not to
OK corrected
Line 408 a lot = many
OK corrected

Line 487, little = small

OK corrected

Line 500, X value contrasts

OK corrected

Line 523 deduce not deduced

OK corrected

Line 525 suggested rather than evidence

OK corrected

Line 526-7 implied by rather than well witnessed by

OK corrected

Line 533 – define long (temporal? Spatial?)

Long-term

Line 536 – trends in not trend of /

OK corrected

Line 550 precipitation not precipitations

OK corrected

Line 559 – alternations between (rather than alternance)

OK corrected

Line 560 glacier not glaciers

OK corrected

Line 574 'the to first order' – does not make sense – reword

Line 584 you mean 'climatic conditions similar to the. . ..'?

Yes. Corrected

Line 608 'The present drainage of the . . ..'

OK corrected

Line 637 helps with (not for)

Sentence deleted

Line 647 Then ? – not needed, remove

Sentence deleted

Line 655 – record (not recorded)

Sentence deleted

Line 658 'was open to the Atlantic Ocean from the. . ..''

OK corrected

Line 659 records not record

OK corrected

[revised manuscript text omitted]

Figure 4

[Figure]

Figure 5

[Figure]

Figure 6

[Figure]

Figure 7

[Figure]

Figure 8

[Figure]

Figure 9

[Figure]

Figure 10

[Figure]

Figure 11

---

## Author Response (AR2)

To the Associated Editor of ESurf

Dear V. Vanacker

Thank you for your last comments on the revised version of our manuscript "Drainage reorganization and divide migration induced by the excavation of the Ebro basin (NE Spain)".

We are pleased to resubmit a revised version of this manuscript

In the following we respond to the three main remaining questions you addressed as well as to the minor comments.

Sincerely yours,

Stéphane Bonnet on behalf of the co-authors

1) In section 3.3, you use a stream power method to analyse differences in specific stream power between the present-day and ancient Duero river (L380-394). This is somehow surprising, as you state in section 3.4 that the results of stream power analyses can be biased by the quality of the topographic data - slope measurement on low-resolution DTM data. Is your argument in section 3.3 robust, or can you rephrase this argument based on the results of the X-analyses?

This is actually a problem of spatial scale of investigation. We first applied a stream power analysis to the Duero profile to investigate large-scale morphology of this river. Here the DEM resolution is sufficient to capture these large-scale features. On the opposite DEM resolution matters for the analysis of the landscape morphology near the Ebro-Duero divide, because we investigated here very detailed spatial variations in morphology that can be biased by the DEM resolution. That is the reason why we applied a X-analysis, which is not sensitive to DEM resolution through slope measurement (Perron and Royden, 2012), rather than a slope-area analysis. We agree that it was potentially confusing in the previous manuscript.

The X-analysis is now frequently used in the community of Earth Surface Processes to infer landscape disequilibrium and divide migration. Actually the reviewers asked us to reduce the presentation of this method in the revised ms. Then we do not think necessary to justify for its use in our study and we modified the ms accordingly:

Prevous ms (lines 400-408):

*The comparison of the shape of longitudinal profiles of rivers across divide is a way that has been proposed recently to infer disequilibrium between rivers and the potential migration of their divide (Willett et al., 2014). Although the slope-area analysis of channel profiles (e.g. Whipple and Tucker, 1999; Kirby and Whipple, 2012) is potentially a powerful tool to evidence differences in the equilibrium state of rivers across divide, and then to infer their migration (Willett et al., 2014), this method is limited and even biased by the quality of the topographic data. Indeed, both a low-resolution of the DEM and corrections brought to the DEM (filling or carving), lead to substantial uncertainties that are automatically transferred to the slope-area data. To avoid slope measurements, Perron and Royden (2012) proposed a procedure based on a coordinate transformation allowing linearizing river profiles…*

New ms (lines 399-404)

*The comparison of the shape of longitudinal profiles of rivers across divide is a way that has been proposed recently to infer disequilibrium between rivers and the potential migration of their divide (Willett et al., 2014).  The χ-analysis of river profiles (Perron and Royden, 2012) is a powerful tool to evidence differences in the equilibrium state of rivers across divide, and then to infer their migration (Willett et al., 2014).  This method is based on a coordinate transformation allowing linearizing river profiles (Perron and Royden, 2012)…*

2) In the discussion, you comment on the link between tectonic activity and landscape evolution. Given that the landscape response time can be (very) long in these systems, can we make a firm statement yet on the drivers of landscape incision (tectonic vs. climate patterns: L501-503)?

At the scale of Northern Iberia, the main driver of incision was likely the change from endorheic to exorheic drainage condition (lines 61-67; lines 102-109; lines 113-114; section 4.2). We extensively discuss this phenomenon in section 4.2. On lines 501-503 (section 4.1; now lines 498-500), we discard the role of local tectonic activity on the pattern of migration of divide that we document. Then, we do not think necessary to add any statement here, as the role of the change from endorheic to exorheic condition in driving incision is further discussed in the next section (4.2)

3) The climate pattern in the wider region is studied based on modern precipitation data (Fig. 6). How relevant are such maps when we look at long-term processes, certainly when considering the climate changes mentioned in L181-206? A short discussion on the relevance of the data for reconstructing past climate change would be welcome - for example L505-519.

On lines 505-519 (now 502-518), we report that our analysis shows that there is still a landscape disequilibrium along the Ebro-Duero divide and that in the absence of modern precipitation contrast there, we conclude that the present divide migration is unlikely driven by difference in climate:

Lines 511-513 of the former revised ms (now lines 508-510) : *"we suggest that the **present day** climatic condition is unlikely to control the general pattern of **current** drainage reorganization between the Ebro and Duero basins"*.

Proxies (section 2.4, lines 181-206) indicate past climatic variations in northern Iberia but they do not allow to infer past precipitation differences along the Ebro-Duero divide. We then add the following sentence in the new ms (Lines 512-514):

*(Lines 512-514) "Existing paleoclimate proxies do not allow to evidence past precipitation differences along the divide that could explain the reorganization of the drainage there."* Moreover, there is no clear evidence of important glacier development and related …

Some detailed comments
L44-45: "drop in the location of drainage divide". Do you mean "elevation"?

No, spatial position: "location" changed to "spatial position"

L50: Would you characterise the X-method as an analytical approach? Check, and rephrase - if necessary
Yes
L63: Would be good to have a reference here
OK -> Riba et al., 1983; Garcia-Castellanos et al., 2003

L72-73: Rephrase sentence

*"Then, these two adjacent basins are characterized by contrasting preservation of their endorheic stages and represent an ideal natural laboratory to evaluate the mechanisms that caused differential post-orogenic incision at the origin of divide migration"*
Rephrased (Lines 71-73):
*"Then, these two adjacent basins are characterized by differences in incision and in the preservation of their endorheic stages. They thus represent an ideal natural laboratory to evaluate divide migration in response to differential post-orogenic incision."*

L175-180: The precipitation map is relevant to study modern precipitation patterns (Fig. 6). What about long-term patterns? Are they expected to be steady over this time period?

As modern precipitation does not who significant contrasts along the Duero-Ebro divide (except in the Cameros area as discussed), it is unlikely that some major contrasts exited in the past

Line 34, now line 28: "*Chi-analysis of river profiles*" changed to "*χ-analysis of river profiles*"

Line63: *χ-method for analyzing longitudinal profiles of rivers*" changed to " *χ-analysis of longitudinal profiles…*"

Lines 234 and 429: "*χ Analysis*" changed to "*χ-analysis*"

Lines 482, 572: "*χ analysis*" changed to "*χ-analysis*"

"*They pointed out several chronostratigraphic evidence that allow them to build a relative chronology of capture events in the Jalon network history.*"

Changed (lines 245-246:

"*They used chronostratigraphic evidence to build a relative chronology of capture events in the Jalon area .*"

"*However despite a similar bedrock we cannot ruled out some local influence of the lithology on the shape of these knickpoints*"

Changed (276-277):

"*However we cannot ruled out some local influence of the lithology on the shape of these knickpoints*"

"*χ predictions*" changed to ""*χ evidence*"

The text has been moved as requested.